# Microbial decomposition of biodegradable plastics on the deep-sea floor

Taku Omura[1], Noriyuki Isobe [2], Takamasa Miura[3], Shun'ichi Ishii [4], Mihoko Mori[3], Yoshiyuki Ishitani[4], Satoshi Kimura[1], Kohei Hidaka [3], Katsuya Komiyama[1], Miwa Suzuki[5], Ken-ichi Kasuya [5,6], Hidetaka Nomaki[4], Ryota Nakajima[7], Masashi Tsuchiya[7], Shinsuke Kawagucci [7], Hiroyuki Mori[8], Atsuyoshi Nakayama[9], Masao Kunioka[10], Kei Kamino[3] & Tadahisa Iwata [1] ✉

Microbes can decompose biodegradable plastics on land, rivers and seashore. However, it is unclear whether deep-sea microbes can degrade biodegradable plastics in the extreme environmental conditions of the seafloor. Here, we report microbial decomposition of representative biodegradable plastics (polyhydroxyalkanoates, biodegradable polyesters, and polysaccharide esters) at diverse deep-sea floor locations ranging in depth from 757 to 5552 m. The degradation of samples was evaluated in terms of weight loss, reduction in material thickness, and surface morphological changes. Poly(L-lactic acid) did not degrade at either shore or deep-sea sites, while other biodegradable polyesters, polyhydroxyalkanoates, and polysaccharide esters were degraded. The rate of degradation slowed with water depth. We analysed the plastic-associated microbial communities by 16S rRNA gene amplicon sequencing and metagenomics. Several dominant microorganisms carried genes potentially encoding plastic-degrading enzymes such as polyhydroxyalkanoate depolymerases and cutinases/polyesterases. Analysis of available metagenomic datasets indicated that these microorganisms are present in other deep-sea locations. Our results confirm that biodegradable plastics can be degraded by the action of microorganisms on the deep-sea floor, although with much less efficiency than in coastal settings.

Annually, 400 million tons of plastic are produced worldwide and used in various industrial and consumer applications to sustain the convenience of modern life. Plastic products should be collected and recycled after use; however, it has been reported that ~8 million tons of plastic waste end up in the marine environment through the rivers every year[1–4]. Once released, plastic debris is assumed to remain in the marine environment for hundreds of years or longer because of its durable, enzyme-resistant chemical composition[5–7]. Plastic debris is carried by waves and currents to the beach or sea surface[8]. Over time, some of the plastic debris sinks to the deep-sea as it becomes heavier than seawater because of biofouling or the accumulation of sand grains on the surface of the plastic[9]. Recently, studies have reported

that the deep-sea floor is a major sink for plastic debris, and large amounts of plastic debris are present on the deep-sea floor; half of these are single-use plastics[10–12].

Biodegradable plastics represent one approach to reducing the proliferation of plastic waste on the deep-sea floor. Approximately 1.14 million tons of biodegradable plastics were produced in 2022[13]. Their biodegradability has been investigated using International Organization for Standardization testing methods in aerobic and anaerobic conditions in compost, soil, and river water[14,15]. In the case of marine biodegradation, a Biochemical Oxygen Demand (BOD) biodegradation test using seawater[15,16] and a field test at the shore[17,18] are performed. However, little is known about the capacity of deep-sea bacterial

communities that live in extreme environmental conditions of temperature and pressure to decompose the most common forms of biodegradable plastic debris on an ecological time scale. In addition, it is not fully understood whether the biodegradable plastics that have been developed currently degrade in this extreme environment, the deep-sea floor, in the same way, that they degrade on shore, or how long it takes for them to degrade.

We addressed this deficit by conducting short- and long-term biodegradation tests at five deep-sea floor locations in Pacific Ocean: three bathyal sites [off Misaki Port (BMS, depth = 757 m), off

Hatsushima Island (BHT, depth = 855 m), and Myojin Knoll (BMJ, depth = 1292 m)], and two abyssal sites [Kuroshio Extension Observatory (AKR, depth = 5503 m) and Minamitorishima Island (AMN, depth = 5552 m)] (Fig. 1a, b and Table 1). At the same time, a control experiment was conducted at the port of JAMSTEC Yokosuka Headquarters (PJM, depth = 2–6 m). Table 1 shows the sites where the samples were placed, latitude/longitude information, depth below sea level, salinity, temperature, dissolved oxygen, date of placement, date of recovery, and an abbreviation describing the site and duration of the placement. All of the samples used in this study (formal names,

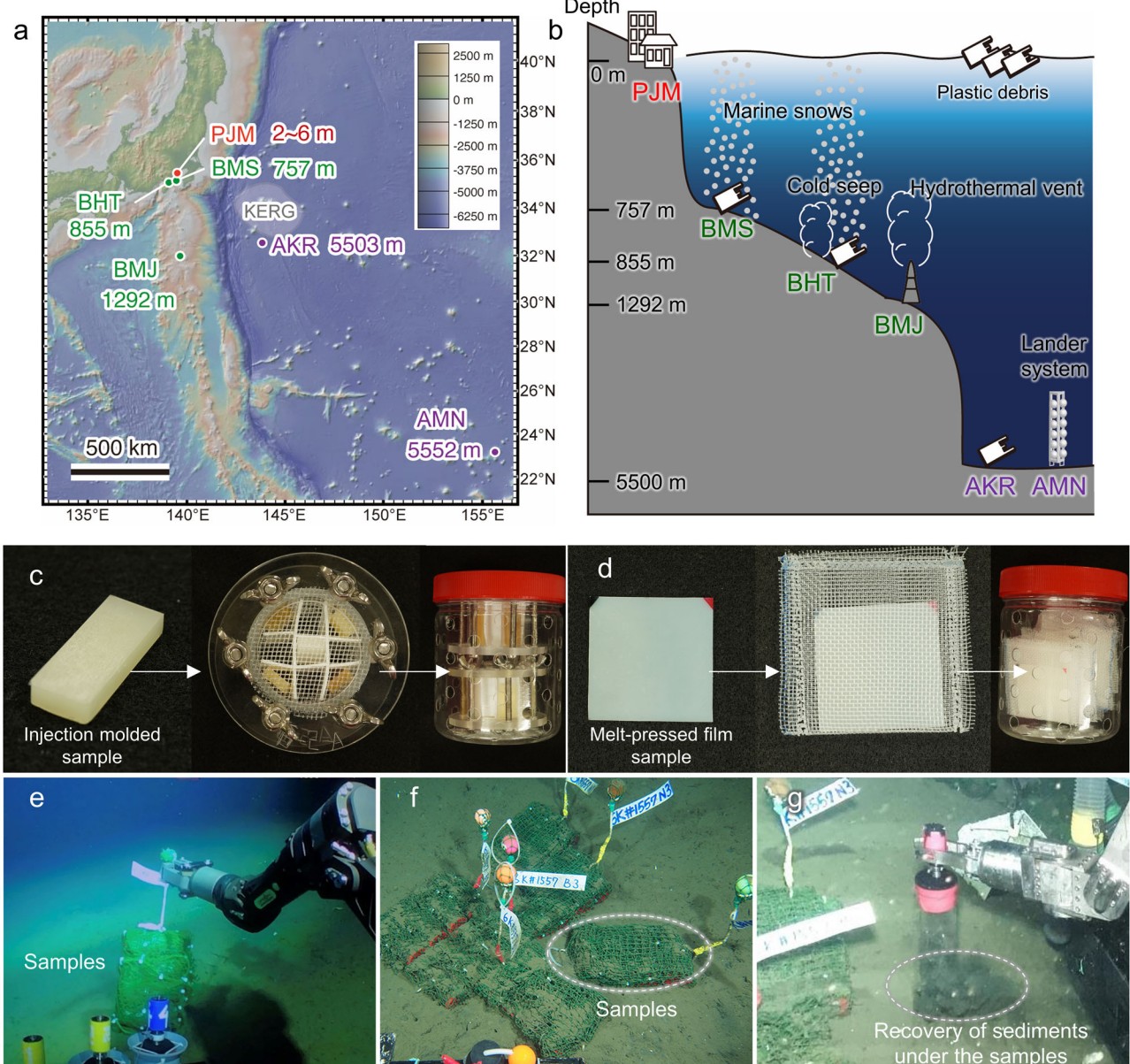

**Fig. 1 | Sample deployment at the deep-sea floor. a** Map of in situ biodegradation test sites. The map was made with GeoMapApp (www.geomapapp.org) / CC BY / CC BY (Ryan et al., 2009)[38]. See Table 1 for abbreviations regarding location. **b** BMS and BHT are close to heavily populated cities and shipping routes to/from Tokyo Bay with a high accumulation of observed plastic debris. BHT and BMJ are sites containing unique chemosynthetic communities near cold seepage and hydrothermal vents, respectively. AKR is south of the Kuroshio Extension recirculation gyre (KERG), where a high accumulation of plastic debris has been reported. AMN is another abyssal environment, but distant from populated areas. PJM is a coastal environment (Port of JAMSTEC Yokosuka Headquarters), and serves as a reference site that is close to large cities and rivers. The sample sets were deployed on the deep-sea floor by the human-occupied submersible Shinkai 6500, except for AMN, where the deployment was performed by a lander system, and the sample sets were placed 2 m above the sea floor. **c** Injection-molded sample specimen used in this study and sample set in a custom-made sample holder (closed with 1 mm gap polyethylene (PE) mesh). **d** Melt-pressed sample specimen used in this study and the sample set with its sample sealed with 1 mm gap PE mesh. **e** The test samples deployed at BHT. **f** The test samples at BHT 4 months after deployment. **g** Recovery of sediments under the samples.

**Table 1 | Summary of the on-site degradation test site**

| Site | Abbr. | Latitude N | Longitude E | Depth m | Salinity | Temp. °C | DO mg/L | Deployment yyyy/mm/dd | Recovery yyyy/mm/dd | Period Days | Abbr. | Sample |
|---|---|---|---|---|---|---|---|---|---|---|---|---|
| Port of JAMSTEC Yokosuka Headquarters | PJM | 35°19.18 | 139°39.05 | 2-6 | 30.0 | 25.5 | 5.7 | 2019/09/30 | 2020/10/03 | 369 | PJM12 | Injection molded |
| | | | | 2-6 | 30.3 | 25.4 | 5.0 | 2021/08/08 | 2021/10/08 | 61 | PJM02 | Film |
| Bathyal hydrocarbon seepage off Hatsushima island | BHT | 35°0.95 | 139°13.33 | 855 | 34.5 | 3.6 | 1.8 | 2019/09/12 | 2020/01/07 | 117 | BHT04 | Injection molded |
| | | | | | | | | 2019/09/12 | 2020/11/16 | 431 | BHT14 | Injection molded |
| | | | | | | | | 2021/02/05 | 2021/05/11 | 95 | BHT03 | Film |
| | | | | | | | | 2021/02/05 | 2021/10/13 | 250 | BHT08 | Film |
| Bathyal seafloor off Misaki port | BMS | 35°4.20 | 139°32.50 | 757 | 34.3 | 4.4 | 2.1 | 2021/05/12 | 2021/10/12 | 153 | BMS05 | Film |
| Bathyal hydrothermal vent in Myojin knoll | BMJ | 32°6.30 | 139°52.17 | 1292 | 34.3 | 4.6 | 2.1 | 2021/05/21 | 2021/10/07 | 139 | BMJ05 | Film |
| Abyssal plain near Kuroshio extension observatory | AKR | 32°34.78 | 143°46.14 | 5503 | 34.7 | 1.6 | 5.2 | 2021/05/20 | 2021/10/06 | 139 | AKR05 | Film |
| Abyssal plain around Minamitor-ishima island | AMN | 22°59.93 | 154°24.55 | 5552 | 34.7 | 1.5 | 5.0 | 2020/03/15 | 2021/04/09 | 390 | AMN13 | Injection molded |

**Table 2 | Formal name, abbreviation in the paper, and thermal properties of samples**

| Category | Formal name | Abbr. | Supply company | $T_g$/°C | $T_m$/°C |
|---|---|---|---|---|---|
| Polyhydroxyalkanoates (PHA) | Poly[(R)-3-hydroxybutyrate] | PHB | Imperial Chemical Industries | 4 | 175 |
| | Poly[(R)-3-hydroxybutyrate-co-12 mol%-(R)-3-hydroxyvalerate] | PHBV | Imperial Chemical Industries | 0 | 150 |
| | Poly[(R)-3-hydroxybutyrate-co-9 mol%-(R)–3-hydroxyhexanoate] | PHBH | Kaneka Corporation | −3 | 110, 133 |
| | Poly[(R)–3-hydroxybutyrate-co-16 mol%-4-hydroxybutyrate] | P3HB4HB | Mitsubishi Gas Chemical Company, Inc. | −7 | 102, 168 |
| Biodegradable polyesters | Poly(butylene succinate) | PBS | Mitsubishi Chemical Corporation | −32 | 115 |
| | Poly(butylene succinate-co-adipate) | PBSA | Mitsubishi Chemical Corporation | −45 | 100 |
| | Poly(ε-caprolactone) | PCL | Sigma-Aldrich | −60 | 60 |
| | Poly(butylene adipate-co-terephthalate) | PBAT | BASF Corporation | −30 | 110 |
| | Poly(ʟ-lactic acid) | PLLA | Shimadzu Corporation | 60 | 170 |
| Polysaccharide ester derivatives | Cellulose (β-1, 4-glucan) | CE | Synthesized by the authors of this work | - | - |
| | Cellulose acetate DS 1.5 | CEA | | - | - |
| | Cellulose triacetate DS 3.0 | CETA | | 170 | 298 |
| | Paramylon (β-1, 3-glucan) | PR | | - | - |
| | Paramylon acetate DS 1.5 | PRA | | - | - |
| | Paramylon triacetate DS 3.0 | PRTA | | 168 | 281 |
| | Mutan (α-1, 3-glucan) | MU | | - | - |
| | Mutan acetate DS 1.5 | MUA | | - | - |
| | Mutan triacetate DS 3.0 | MUTA | | 168 | 338 |
| Non-biodegradable common plastics | Polypropylene | PP | Japan Polypropylene Corporation | 5 | 175 |
| | Low-density Polyethylene | PE | FUJIFILM Wako Pure Chemical Corporation | −120 | 120 |
| | Polyethylene terephthalate | PET | Sigma-Aldrich | 70 | 270 |
| | Polystyrene | PS | FUJIFILM Wako Pure Chemical Corporation | 100 | - |

abbreviations, thermal properties, and chemical structures) are summarized in Table 2 and Supplementary Fig. 1.

Biodegradable plastic-degrading microorganisms have been isolated from compost, soil, rivers, and seawater[19,20]. Degrading enzymes that they secrete have been isolated and purified[21], and biochemical properties, amino acid sequences, and three-dimensional crystal structures of these enzymes have been reported[22]. However, only a few polyhydroxyalkanoate (PHA)-degrading microorganisms have been reported on the deep-sea floor[23], and no microorganisms capable of

degrading other biodegradable plastics are known in that environment, raising questions about deep-sea degradation mechanisms. Therefore, investigating the decomposition of biodegradable plastics on the deep-sea floor will significantly contribute to the development of knowledge and technologies in the fields of materials science and biology, in addition to environmental conservation and understanding of the deep-sea environment. In this study, the decomposition of biodegradable plastics in the deep-sea was investigated for biodegradable polyesters and polysaccharide ester derivatives, in addition

to PHA, which has been actively studied: we assessed weight loss, reduction in material thickness, and surface morphological changes at the deep-sea floor, biofilm formation, microbial accumulation, and candidate genes for degradation of plastics by growing microbes.

## Results and discussion

### Biodegradation at the deep-sea floor

We investigated the decomposition of representative biodegradable plastics (PHA, biodegradable polyesters, and polysaccharide esters) at the above-described five deep-sea floor locations (Fig. 1a, b). Injection-molded samples and films were placed in custom-made sample holders and mesh bags, respectively, and were placed on the deep-sea floor in a condition that prevented physical deformation [samples were placed in polyethylene terephthalate (PET) containers with holes, protected by tennis nets (Fig. 1c, d, Supplementary Figs. 2, 3)]. Installation in the deep-sea and recovery of the samples were performed aboard the *Shinkai 6500* human-occupied vehicle (HOV) using a robotic arm. The seafloor soil immediately below the samples was also recovered using a custom-made core and used for microbiological analysis (Fig. 1e–g, Supplementary Fig. 4).

### Biodegradation of biodegradable polyesters at the deep-sea floor

Figure 2a–c shows the overall shapes and morphology of poly[($R$)-3-hydroxybutyrate-*co*-($R$)-3-hydroxyhexanoate] (PHBH) injection-molded samples placed on shore (PJM12) and off Hatsushima Island (BHT14) and Minamitorishima Island (AMN13) for approximately 1 year. The PHBH sample shown as an example is one in which degradation had progressed relatively well compared with other samples. The photographs taken from the top and the end after ultrasonic washing and drying, revealed that no physical deformation had occurred. Furthermore, it was confirmed by X-ray diffraction that the crystal structure of samples remained unchanged during the submersion periods at the deep-sea floor and experimental processing. Scanning electron microscopy (SEM) images (Fig. 2d) show the surface profiles after removal of microorganisms from the surface. Whereas the surface morphology before degradation (original) was very smooth, the surfaces of the samples after placement on the shore or the deep-sea floor were observed to be uneven, and degradation was in progress. This kind of degradation that creates unevenness on the surface of the material is considered to be due to microbial degradation, not physical deformation or chemical degradation. The fact that degradation progresses from the surface homogeneously is evident from the extreme decrease in the thickness of the samples from the shore (PJM12). In about 1 year, the thickness of the samples, which was initially 4000 μm, was found to have decreased by ~700 μm at the shore (PJM12), ~110 μm off Hatsushima Island (BHT14), and ~10 μm at Minamitorishima Island (AMN13). On the basis of this series of observations, the degradation was considered to be microbial degradation, rather than physical deformation.

Figure 2e–g shows the shapes, stereomicroscopic images, and SEM images of surfaces of PHBH film samples placed off Hatsushima Island for 8 months (BHT08), respectively, as an example of film degradation at the deep-sea floor. In addition, residual weight and film thickness after 3 and 8 months are shown in Fig. 2h. In all samples, no edges were chipped, and no physical deformation occurred during submersion at the deep-sea floor or during subsequent ultrasonic washing. The original PHBH sample films were slightly opaque with very smooth surfaces (Fig. 2e, Original). After about 8 months submerged off Hatsushima Island, microorganisms adhered to the surface, and a yellowish, slimy appearance was observed (Fig. 2e, BHT08). When the film was fixed with formaldehyde and SEM observation was performed, it was observed that a large number of microorganisms adhered to the film surface, forming a biofilm (Fig. 2g, BHT08). When the film was ultrasonically washed and dried, and then observed under

a stereomicroscope, numerous holes with diameters of ~50 μm were observed (Fig. 2f), which appeared to have been created by the degradation of the film. SEM observation revealed a very large number of spherical holes and unevenness [Fig. 2g, BHT08 (after washing)]. These are thought to have been created by the degradation of the film by microorganisms, because this degradation pattern, such as spherical holes on the film surface observed by SEM, is quite similar to that in a previous report where PHB film was degraded by microorganisms isolated from soil[24].

Figure 2h (Supplementary Data 1, Supplementary Tables 1 and 2) shows the change in residual weight and thickness of the PHBH film. It was found that film weight decreased by 22% in 3 months and 52% in 8 months, and film thickness decreased by 35 μm in 3 months and 70 μm in 8 months from 225 μm initial thickness. Furthermore, the residual weight decreased linearly, suggesting that degradation occurred at a constant rate over the 8-month period. The fact that the slope of the residual weight change and the slope of the film thickness decrease were almost the same suggests that degradation in the film thickness direction is dominant.

Figure 2i shows schematically the microbial degradation of biodegradable plastic. Microorganisms attach to the surface of the plastic film and form a biofilm. Some of the microorganisms secrete degradative enzymes that cleave molecular chains on the film surface and break them down into water-soluble monomers or oligomers. The film surface is then degraded, and the film becomes uneven. This series of microbial degradations is basically the same as those in soil and rivers. Whether or not the water-soluble monomers and oligomers were completely degraded to carbon dioxide and water by microorganisms in seawater was confirmed by a BOD biodegradation test using seawater in a laboratory setting. Figure 2j (Supplementary Data 2) shows the results of the BOD biodegradation test of PHBH and reference material (cellulose). It was confirmed that they were degraded to carbon dioxide and water in about 1 month. Considering the above results comprehensively, at deep-sea floors or shore, biodegradable plastics undergo degradation from the surface by microorganisms, forming water-soluble intermediates, which are finally metabolized to carbon dioxide and water. However, in this study, no further experiments were conducted to directly support that microorganisms actually secrete extracellular depolymerases in the extreme environment of the deep-sea. Subsequently, it is unclear whether the microorganisms at deep-sea completely degrade biodegradable plastics into carbon dioxide and water through water-soluble intermediates.

Figure 3a–c (Supplementary Data 1, Supplementary Tables 1 and 2) shows the changes in film morphology, surface SEM images, residual weight, and film thickness for poly(butylene succinate-*co*-adipate) (PBSA) and poly(L-lactic acid) (PLLA), which are biodegradable polyesters, and polypropylene (PP), a non-biodegradable common plastic, respectively, which were placed off Hatsushima for 3 and 8 months. The surface SEM image of PBSA also showed the formation of biofilm on the surface. In the SEM image of the surface, unevenness and holes were observed, indicating that the surface had been decomposed by microorganisms. Therefore, it is considered that microbial degradation of PBSA in the deep-sea progresses in the same manner as that of PHBH (Fig. 2e–g). However, the weight loss rate and changes in film thickness were much slower/lower than those for PHBH.

In the case of PLLA, microorganisms were observed adhering to the film surface, but after 8 months, the film surface was very smooth, indicating that no degradation had occurred, even though PLLA is categorized as a biodegradable polyester (Fig. 3b). The sample weight and thickness remained unchanged during submersion at the deep-sea floor. This is thought to be due to the absence of bacteria capable of degrading PLLA among the attached microorganisms. PLLA clearly does not degrade at the deep-sea floor. Nor does it degrade in soil and rivers—it is well known that PLLA is only degraded in compost

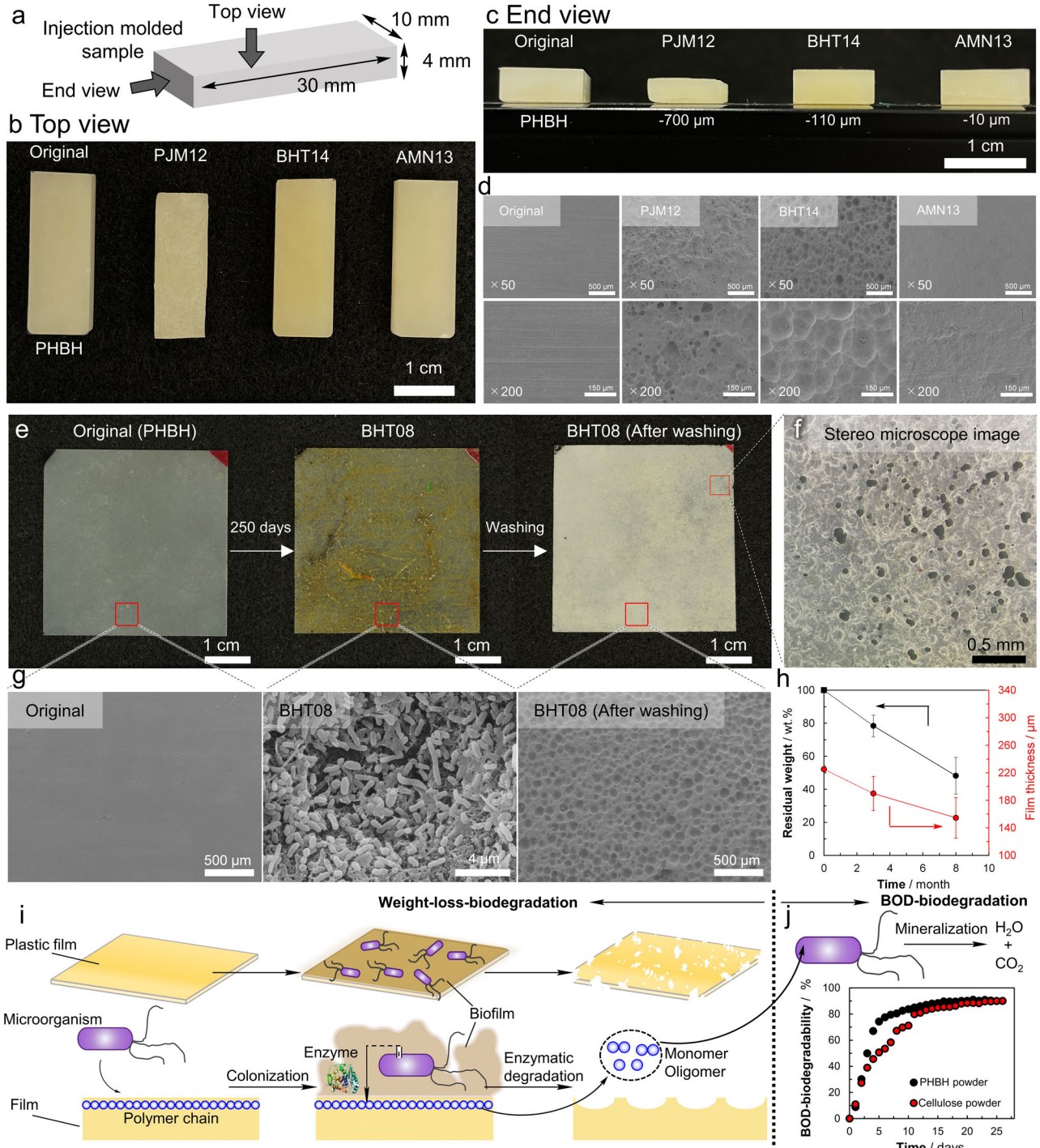

**Fig. 2 | Biodegradation of Poly[(R)-3-hydroxybutyrate-co-9 mol%-(R)−3-hydroxyhexanoate] (PHBH) injection-molded and melt-pressed film samples placed at the deep-sea floor and models of microbial degradation. a** Injection-molded samples were placed in the sea at different depths (2–5500 m) for about 1 year for biodegradation tests. See Table 1 for abbreviations regarding location and Table 2 for abbreviations regarding samples. **b, c** The original and PJM12, BHT14, and AMN13 injection-molded samples are shown in a photograph taken from (**b**) the top side and (**c**) the end, showing that the samples were reduced in size while maintaining their shape without physical collapse. **d** Surface SEM images show that decomposition proceeded regardless of water depth. **e** Photographs of an original film sample of 4 × 4 cm, after 250 days of placement on the deep-sea floor off Hatsushima Island at 855 m (BHT08), and after biofilm removal by ultrasonic washing, respectively. **f** The dimensions of the film in BHT08 (after washing) were the same as the original, meaning that no physical collapse occurred, but numerous penetrating holes were observed by stereomicroscopy. **g** Scanning electron

microscope (SEM) images showing a magnified view of the surface of each PHBH film. While the surface of the original sample was smooth, microorganisms accumulated on the surface of the BHT08 sample, and the surface of the film, after the microorganisms were removed by ultrasonic washing (BHT08 After washing), showed the roughness typically observed following microbial degradation. **h** The decrease in weight (black line) and film thickness (red line) at the deep-sea bottom off Hatsushima Island over 8 months. Values are given as average of $n = 4$ independent samples with its standard deviation (Supplementary Data 1). **i** Illustrations of biodegradation corresponding to the photos and SEM images above. **j** BOD biodegradability curve, showing that PHBH (black dot) and cellulose (red dot) are completely converted to water and carbon dioxide by microorganisms in the seawater of Tokyo Bay. The micrographs are representative images from $n = 3$ independent samples with similar results. The BOD data is representative of one of $n = 3$, with similar results (Supplementary Data 2). Source data are provided as a Source Data file.

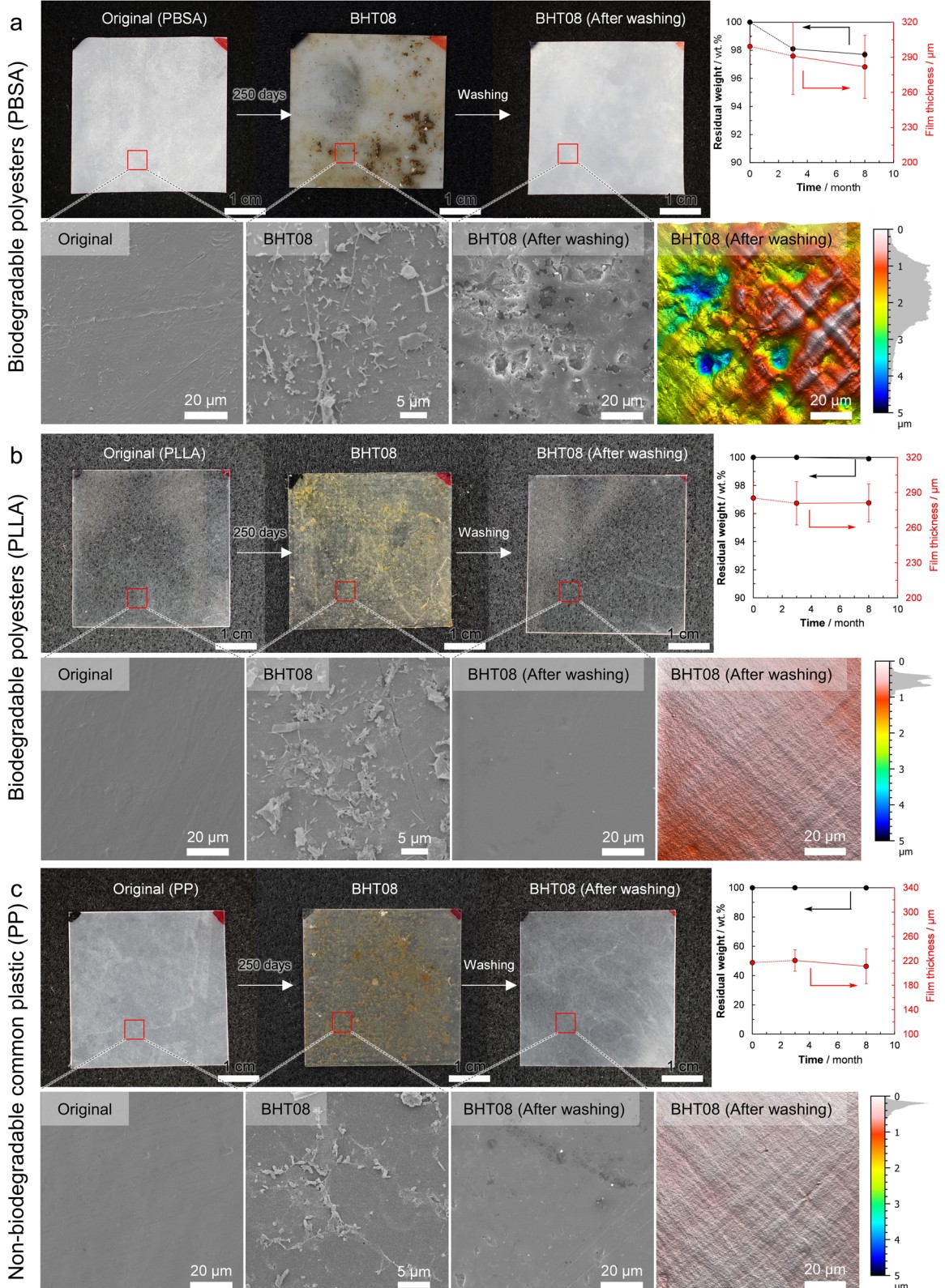

**Fig. 3 | Analysis of Poly(butylene succinate-*co*-adipate) (PBSA), Poly(ʟ-lactic acid) (PLLA), and Polypropylene (PP) melt-pressed film samples placed at the deep-sea floor (BHT).** The degradability of **a** PBSA film, **b** PLLA film, and **c** PP film at the deep-sea floor. For PBSA film, microorganisms were observed on the film surface after 250 days of placement on the deep-sea floor off Hatsushima Island at 855 m (BHT08), and roughness on the film surface after biofilm removal, slight weight loss, and decrease in film thickness was also observed. For PLLA and PP, few microorganisms were observed on the film surface even after 8 months in BHT, and the film surface remained smooth after ultrasonic washing, with no observed weight loss or decrease in film thickness. Values are given as average of $n = 4$ for PBSA and PLLA, $n = 2$ for PP independent samples with its standard deviation (Supplementary Data 1). The micrographs are representative images from $n = 3$ independent samples with similar results. Source data are provided as a Source Data file.

(temperature: >60 °C, humidity: >60%). Therefore, it is desirable to collect PLLA after use and actively promote its decomposition by composting.

In the case of PP, which is a representative, common, non-biodegradable plastic, it was observed that the amount of micro-organisms attached to the film surface was low (Fig. 3c). This may be because the surface of PP is hydrophobic, unlike biodegradable polyesters, which have hydrophilic ester groups. It was clear that the PP did not degrade at the deep-sea floor.

Results for other biodegradable polyesters and non-biodegradable common plastics after 8 months are shown in Supplementary Figs. 5–9; those after 3 months are shown in Supplementary Figs. 10–12. The SEM images after 3 and 8 months suggest that microbial degradation was in progress for the other biodegradable polyesters (PHB, PHBV, P3HB4HB, PCL, PBS, PBAT), like for PHBH and PBSA. However, no degradation occurred in the other non-biodegradable common plastics (PE, PS, PET), like for PP and PLLA (see Table 2 for definitions of sample abbreviations).

The weight changes of injection-molded samples and films were measured; the degradation rate calculated from the surface area and the submersion periods was defined as the biodegradation rate (Fig. 4a, b, Supplementary Data 1, 3, Supplementary Tables 1–6).

The biodegradation rates of injection-molded samples of PHAs and biodegradable polyesters submerged at BHT for 431 days (BHT14), AMN for 390 days (AMN13), and PJM for 369 days (PJM12) are shown in Fig. 4a and Supplementary Table 3. The biodegradation rate in the marine environment was calculated by normalizing the biodegradation weight loss with respect to initial surface area per day (μg/cm²/day). It was confirmed that PLLA and four types of non-biodegradable common plastic (PE, PP, PS, PET) did not degrade at all in either the shore or deep-sea sites, while the biodegradable polyesters except for PLLA were found to degrade to varying degrees. The biodegradation rate of PHA at the shore in this study was similar to previously reported results at shores worldwide[25]. Interestingly, comparing the biodegradation rate of PHA at the deep-sea sites with that on shore, the rate tended to slow down with depth.

The biodegradation rates of films over a short period of time at other sites and at different depths are shown in Fig. 4b and in Supplementary Tables 1, 2, 4–6. Interestingly, a clear difference in biodegradation rate was observed between the BHT, BMS, and BMJ sites at depths of ~1000 m compared with the AKR and AMN sites at depths of ~5000 m. For PHB, which had the slowest decomposition rate among PHAs, compared with the decomposition rate of 107 μg/cm²/day at the shore (PJM02), at a depth of ~1000 m (BHT03, BMS05, and BMJ05), the decomposition rate was 10.9–19.8 μg/cm²/day, and at a depth of ~5000 m (AKR05 and AMN13), the decomposition rate was 4.7–5.5 μg/cm²/day. At ~1000 m, the decomposition rates at all sites (BHT, BMS, and BMJ sites) were similar, even though BMJ is a hydrothermal vent. It is likely that similar results were observed at BMJ to these other sites because the seawater is heated by the hydrothermal fluid (>270 °C); however, even in the immediate vicinity of the hydrothermal vents, the seawater is cooled quickly to 4.6 °C by cold water flow[25]. Our results comparing the biodegradation rate of biodegradable plastics at the shore and the deep-sea floor, where large amounts of plastics are thought to have sunk, reveal that the rate of biodegradation at the deep-sea floor is slower than at the shore, and that the biodegradation rate depends on the depth (Fig. 4c, Supplementary Data 1). Evidently, the result that the biodegradation rate depends on the depth is related to the local number and diversity of degrading microorganisms, which will be discussed later (Fig. 5a, b), in addition to the fact that the deep-sea is an extreme environment of particularly low temperature, high water pressure, and no sunlight.

Using the degradation rates of biodegradable plastics at the deep-sea floor obtained in this study, we calculated the time to complete biodegradation of plastic bags at the shore and in the deep-sea off Hatsushima (BHT), assuming that plastic bags of 15 μm thickness were made of four different PHAs. From the experimental results, PHA degrades on average by about 35 μm of thickness in 3 months, and thus 15 μm of degradation can occur well within the experimental period. At the shore, the calculated degradation time was 6 days for PHBH, which was considered to be the fastest-degrading of the tested PHAs, and 13 days for PHB, which was the slowest-degrading (Fig. 4d). These values are almost the same as those previously reported[26]. At the deep-sea floor, the biodegradation periods were calculated to be about 19 days for PHBH and 58 days for PHB (Fig. 4d). Whereas non-biodegradable common plastics such as PE and PP are not degraded for hundreds of years, these biodegradable plastics will be biodegraded at the deep-sea floor in an acceptable period of time. Therefore, these biodegradable plastics can be considered environmentally friendly materials.

## Biodegradation of polysaccharide esters at the deep-sea floor

Polysaccharides are polymers of biological origin that are abundant all over the world. The structures of polysaccharides consist mainly of glucose units linked by two types of glycosidic linkage (α- and β-), which enable them to perform a variety of functions[27]. For example, cellulose, a natural polymer of glucose bound by β−1,4-glycosidic bonds, provides strength as the main component of plant cell walls. Other examples are paramylon from microalgae, glucomannan isolated from the tubers of *Amorphophallus konjac*, and chitin or chitosan from crustacean shells. In addition, polysaccharides with different glycosidic linkages and molecular structures, including the non-natural polysaccharide mutan linked by α−1,3-glycosidic linkages that we have developed through enzymatic polymerization, have great potential to be useful bioplastics[28].

The weight loss for cellulose (β−1,4-glucan, CE), paramylon (β−1,3-glucan, PR), mutan (α−1,3-glucan, MU), and their acetate derivatives with different degrees of substitution (DS), at the shore and in the deep-sea, are shown in Fig. 4e, Supplementary Table 7 and Supplementary Data 3. All samples were degraded in the deep-sea, although the weight loss was slower than at the shore. For cellulose acetate (CEA with DS = 1.5 and CETA with DS = 3.0), although it has been reported that the degradability changed depending on the degree of DS in activated sludge[29,30], interestingly, the weight loss at the deep-sea floor was similar regardless of the DS. This result suggests that highly-substituted polysaccharide esters with thermoplasticity are degraded by long-term submersion in the deep-sea, perhaps due to de-esterification caused by prevailing alkaline conditions[31]. Paramylon triacetate (PRTA, DS = 3.0) and mutan triacetate (MUTA) were also degraded in the deep-sea, like cellulose triacetate (CETA). These results indicate that polysaccharide esters have great potential as marine-biodegradable plastic materials.

## Microbial community analysis of plastispheres

Plastispheres are defined as microbial communities formed on plastic debris in the ocean[32]. To help determine the mechanism of biodegradation in marine environments, the microbial community structures of deep-sea plastispheres were analyzed by 16S rRNA gene amplicon sequencing. The diversity of microbial communities attached to plastic surfaces was assessed based on amplicon sequence variants (ASVs). The diversity indexes of microorganisms present on biodegradable polyesters decreased with sea depth, and also decreased while biodegradation rates increased (Fig. 5a, b, Supplementary Data 4). A very large number of microorganisms exist in nature. Therefore, in the initial stage, many types of microorganism are expected to adhere to the surface of the plastic. Among them, only a few microorganisms are capable of degrading biodegradable plastic. Over time, microorganisms that can degrade plastic are able to multiply, resulting in a relative increase in the percentage of degrading microorganisms[32,33]. In fact, as shown in Fig. 5a, b, comparing, for

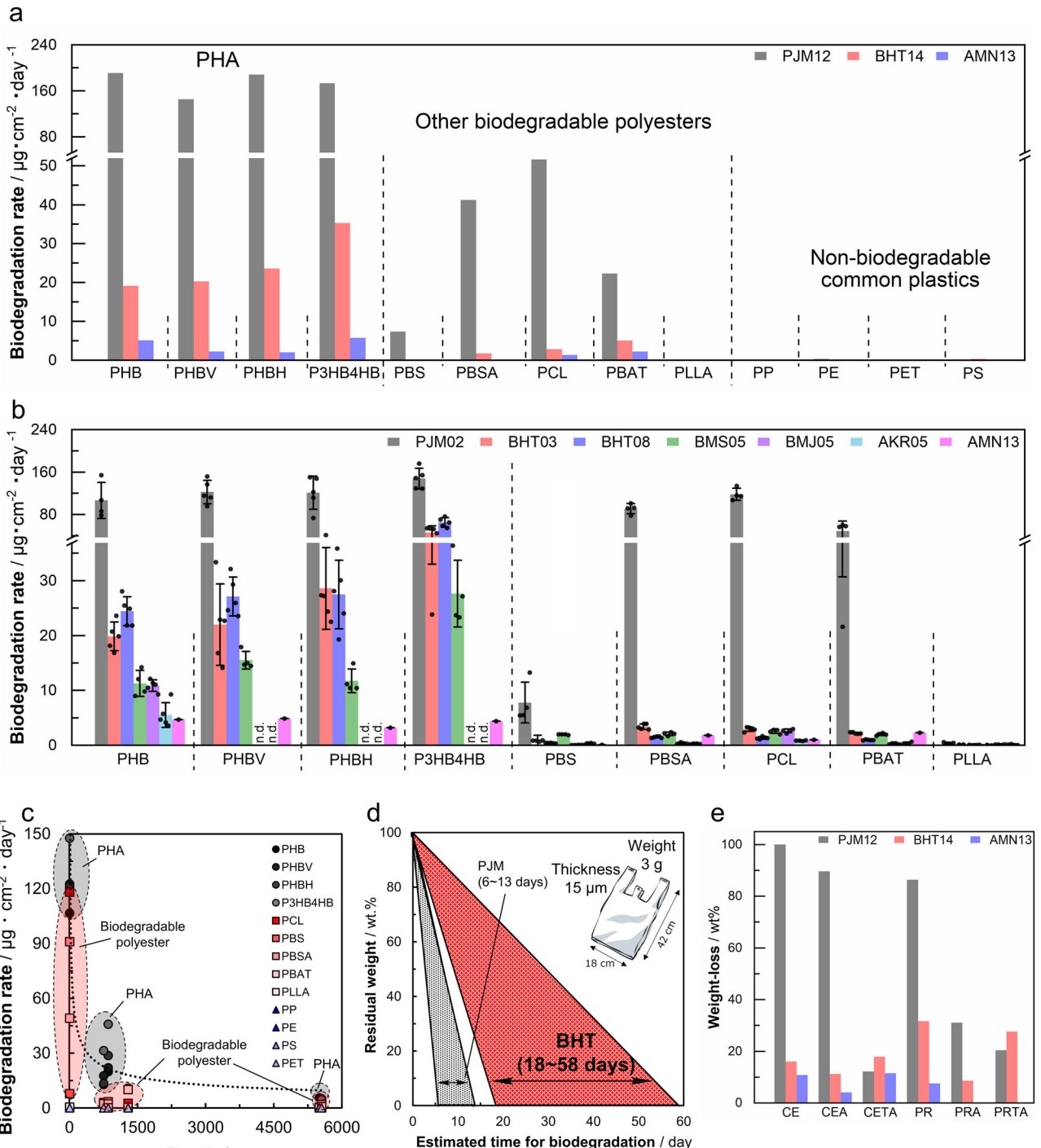

**Fig. 4 | Weight-loss biodegradation rate of plastics in various deep-sea areas and the degradation process for plastic films. a** The weight-loss biodegradation rate of injection-molded plastic samples placed in various ocean environments (2-6 m depth, off the port of JAMSTEC Yokosuka Headquarters; 855 m depth, off a bathyal hydrocarbon seepage near Hatsushima Island; and 5552 m depth, off an abyssal plain around Minamitorishima Island) for ~1 year. Values are given from $n=1$ results (Supplementary Data 3). See Table 1 and Table 2 for abbreviations regarding installation location and samples. **b** The weight-loss biodegradation rates of various melt-pressed PHA and biodegradable polyester films when placed on the shore or at various deep-sea sites. Values are given as average of $n=4$ independent samples with its standard deviation (Supplementary Data 1). **c** Plot of the biodegradation rate of each plastic against depth, showing that the rate slows with depth. The represented data is the average of $n=2$ or $n=4$ independent samples. **d** The estimated period for biodegradation of a typical plastic bag (size: 18 cm × 42 cm, thickness: 15 μm, weight: 3 g) based on the results of this study if it is replaced by the above PHA. **e** The weight-loss of polysaccharide ester derivatives placed in various ocean environments (2-6 m at PJM, 855 m at BHT, 5552 m at AMN) for ~1 year. Values are given from $n=1$ results (Supplementary Data 3). Source data are provided as a Source Data file.

example, BHT04 with BHT14, we can see that the microbial diversity decreased over time (i.e., with the progress of degradation). The diversity of microorganisms on non-biodegradable common plastics, such as PE and PP, did not change across test sites, while PLLA, which is

a compostable plastic, showed the same trend as non-biodegradable common plastics.

In biofilms following several months of submersion at the deep-sea floor, the dominant microorganisms were mainly aerobic

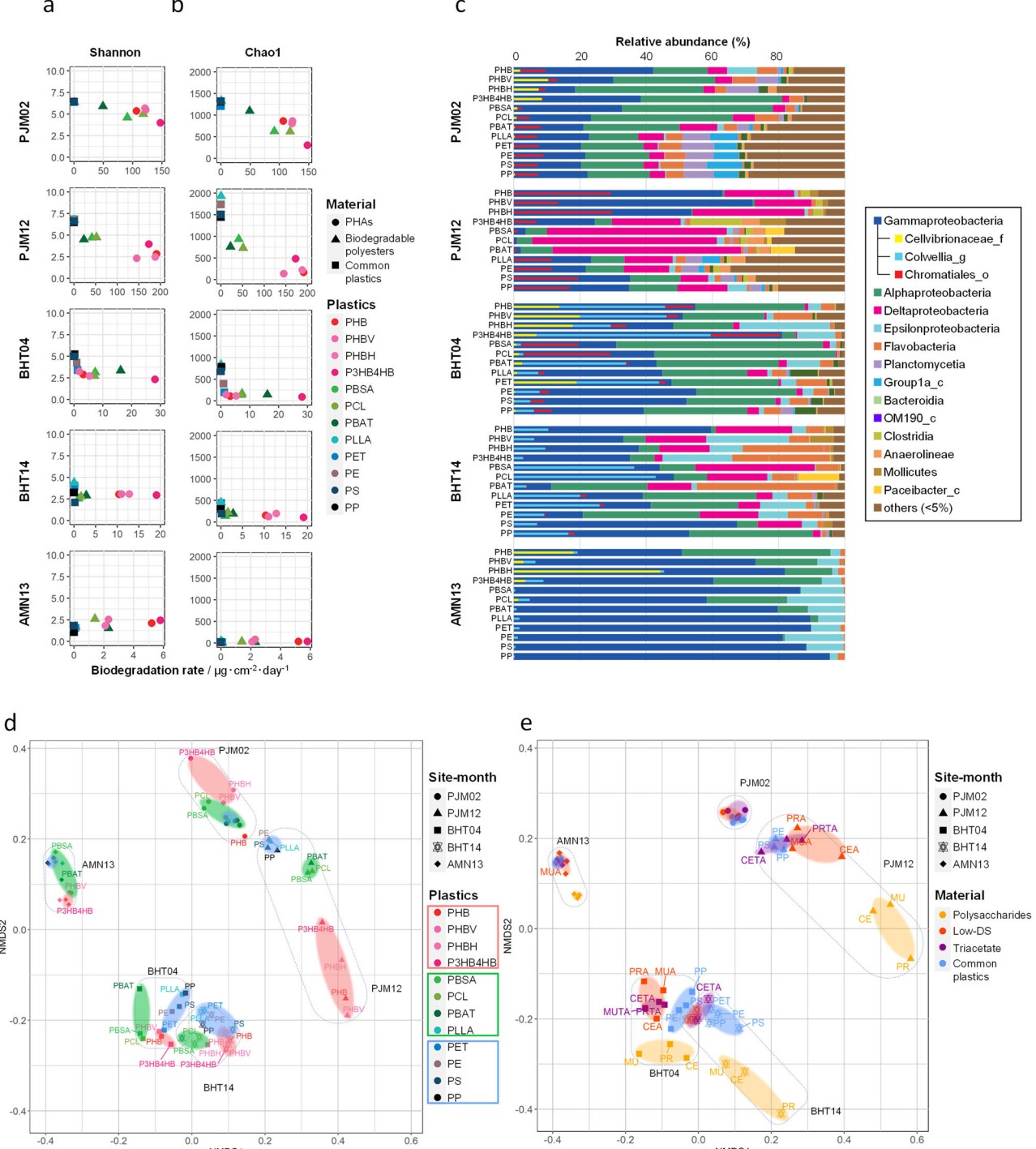

**Fig. 5 | Diversity of microbial assemblages on plastics at three marine sites with different depths, based on 16S rRNA gene amplicon sequencing.**
**a**, **b** Correlation between the diversity of microbial communities associated with plastics in marine environments and the biodegradation rates of the plastics (Supplementary Data 4). Alpha-diversities of the bacterial taxa are indicated by the Shannon (**a**) and Chao1 (**b**) indexes. The biodegradation rates of plastics are shown in μg cm⁻² day⁻¹. Plots are colored to indicate the type of plastic, and the materials are categorized into three groups and marked as follows, circles: PHAs including PHB, PHBV, PHBH, and P3HB4HB; triangles: biodegradable plastics including PLLA, PBSA, PCL, and PBAT; squares: common plastics, including PE, PP, PS, and PET.
**c** Bacterial distribution at the class level on plastics immersed in marine environments. The mean relative abundance (%) of bacterial taxa on each plastic in the

marine immersion test is represented in a horizontal bar chart. Sampling sites and immersion periods are abbreviated as defined in the "Methods" and are denoted on the left (Supplementary Data 4). Amplicon sequence variants belonging to each taxonomic group at the class level are color-coded, as shown to the right. "Others" denotes taxa with relative abundances below the cut-off level of 5%. Inner bars in the Gammaproteobacteria indicate the relative abundances of three families, further characterized by pairwise metagenomic data. **d**, **e** Non-metric multi-dimensional scaling (NMDS) ordination of bacterial communities on PHAs, biodegradable plastics, and common plastics (**d**) and on polysaccharides and their ester derivatives and common plastics (**e**), in marine environments (Supplementary Data 4). NMDS stress value: 0.113421. Source data are provided as a Source Data file.

*Gammaproteobacteria* and *Alphaproteobacteria*, as determined by 16S rRNA gene amplicon sequencing analysis of plastispheres. In mature biofilms, after 1 year at PJM and BHT, the dominant microorganisms were mainly anaerobic *Deltaproteobacteria* (Fig. 5c, Supplementary Fig. 13, Supplementary Data 4). This change might reflect the fact that the sample holders had started to become buried in sediment (Fig. 1f). These trends were similar to those in a metagenome-based community composition analysis based on the ribosomal protein S3 (RpsC) gene, while the redox state of the plastisphere metagenome was shifted to more reductive after long-term submersion at the deep-sea floor (Supplementary Fig. 14, Supplementary Data 5, 6).

Metagenomic analysis of plastispheres in the deep-sea water near and surrounding the sample chamber at BHT, as well as in the sediments below the seafloor (1, 3, 5, and 10 cm under the chamber), was also performed. The results obtained were compared with the microbial communities attached to the surface of biodegradable plastic immersed at BHT for 4 and 14 months (Supplementary Fig. 14, Supplementary Data 5, 6). The microbial population attached to the plastic surface after 14 months became closer to the microbial community in the sediment (1–5 cm), but not close to that in deep-sea water (Supplementary Figs. 15 and 16, Supplementary Data 5, 6). These results indicate that long-term immersion may have affected the plastispheres via the sediment covering the sample chambers and that anaerobic microorganisms such as *Deltaproteobacteria*, "Candidatus *Reidiella*", and others may have attached from the sediment and grown in the plastispheres (Supplementary Fig. 17, Supplementary Data 5, 6).

A non-metric multidimensional scaling (NMDS) plot based on 16S rRNA gene amplicon sequences clearly visualized the category and transition of plastispheres on the 12 plastics tested at the three marine sites PJM, BHT, and AMN (Fig. 5d). Geographical location was the predominant determining factor in the microbial composition of the plastisphere, while submersion time which was associated with a change from an aerobic to an anaerobic environment and the category of plastic were also factors affecting the clustering in the NMDS plot. The plots for plastispheres from the PJM and BHT sites shifted to the lower-right side over time, suggesting that the conditions inside the sample holder may have changed from aerobic to anaerobic. Interestingly, plots for PLLA clustered with the non-biodegradable common plastics rather than the biodegradable polyesters.

In the case of polysaccharide ester derivatives, the overall trends were similar to those for biodegradable polyesters (Fig. 5e, Supplementary Figs. 18 and 19). Three main clusters of plastispheres were observed, and, over time, the positions of these clusters shifted. Interestingly, with an increase in DS, the clusters for polysaccharide esters moved closer to those for the common plastics. Polysaccharide ester derivatives with high DS have a highly hydrophobic surface, similar to that of common plastics, making it more difficult for microorganisms to attach and degrade them.

### Metagenomic analyses identifies potential plastic-degrading microorganisms

Microorganisms that grew specifically on biodegradable plastics were analyzed for the presence of polymer-degrading genes in their genomes using genome-centric metagenomics (Fig. 6a, Supplementary Figs. 14, 20 and 21, Supplementary Table 8, Supplementary Data 5–8). We identified 17 high-quality draft genomes (metagenome-assembled genomes, MAGs) of dominant microorganisms from the metagenomes, and all correlated with highly abundant ASVs based on 16S rRNA gene amplicon sequencing analysis (Fig. 6a, Supplementary Figs. 21–24, Supplementary Data 9). A weighted canonical correspondence analysis (wCCA) plot (Fig. 6b) showed that the genera *Colwellia*, *Agarilytica*, and *Micavibrio* correlated with PHA plastispheres in aerobic conditions, while the order QZLD01 (assigned as "Candidatus *Reidiella*" for RpsC analysis) correlated with PHA plastispheres in

anaerobic conditions. Genes encoding putative PHA-degrading enzymes were identified from microbial species belonging to the classes *Gammaproteobacteria*, *Alphaproteobacteria*, and *Deltaproteobacteria*. Furthermore, genomes of *Colwellia*, *Agarilytica*, and QZLD01 each encoded >10 secretable PHA depolymerases[20], suggesting that marine biodegradation may be caused by these PHA-degrading enzymes (Fig. 6c, Supplementary Figs. 24, 25a, 26–31 Supplementary Table 9, Supplementary Data 9). *Colwellia* is known as a PHA-synthesizing bacterium in the deep-sea[34] and is also observed in the marine plastisphere on PE[35], while *Agarilytica* is known to degrade carbohydrate polymers such as agar[36]. However, there is no report that *Colwellia* and *Agarilytica* secrete a dozen PHB depolymerases on the deep-sea floor. Interestingly, an organism in the order QZLD01 was also detected in the metagenome of plastispheres formed when PHA was submerged in the northern Gulf of Mexico[37] and is hypothesized to biodegrade PHA in the deep-sea around the world, despite remaining uncultivated in the laboratory.

Among the other microorganisms with apparent ability to degrade biodegradable polyester, the genomes of *Profundibacter*, *Hyphomonas* and *Desulfobacula* were each found to encode >2 putative degradation enzymes, secreted polyesterases and cutinases[20] (Fig. 6c). In addition, four of the six MAGs from other microbial species that appeared to lack secreted PHB depolymerases were found to encode >4 putative degradation enzymes (cutinases and polyesterases) (Supplementary Figs. 25a, 26–31, Supplementary Table 9, Supplementary Data 9). This suggests that microorganisms capable of degrading biodegradable polyesters have material specificity. The microorganisms described in this section have not been reported to degrade biodegradable polyesters. However, metagenomic analyses together with microbial community analysis suggest that microorganisms harboring putative degradation genes are predominant on plastic surfaces undergoing degradation in deep-sea sites, and that the biodegradable plastics that showed degradation in this study were marine biodegradable, i.e., the polymer structure can be degraded by the action of microorganisms. To clarify these matters, it will be necessary in the future to isolate microorganisms that appear to be capable of degrading certain biodegradable plastics and to isolate the degrading enzymes they may secrete.

### Global distribution of potential marine plastic-degrading microorganisms

We aimed to generalize the results of this study to worldwide oceans, in consideration of potential regional differences, by surveying the global distribution of microorganisms identified in this study that are potentially involved in the decomposition of biodegradable plastics. The full-length rRNA gene amplicon sequences obtained in silico from the MAGs were used as a reference dataset to compare against publicly available information on marine sediments (Fig. 6d, Supplementary Fig. 25b). The results showed that *Colwellia* (which is associated with the aerobic degradation of PHA) is distributed in marine sediments worldwide, and *Agarilytica* (also associated with PHA degradation) and *Hyphomonas* (associated with degradation of biodegradable polyesters) are also relatively widespread in various deep oceanic regions around the world that harbor plastispheres. Considering the presence of these microbial communities, PHAs and other biodegradable polyesters examined in this study might undergo degradation in other oceanic zones as well.

## Conclusions

This study shows that biodegradable plastics are degraded at the final resting site of oceanic garbage, the deep-sea floor, at different depths, and that specific microorganisms appear to have the capacity to decompose most biodegradable plastics, in particular PHA, biodegradable polyesters, and polysaccharide ester derivatives. The biodegradation rate of biodegradable plastics slows down with depth.

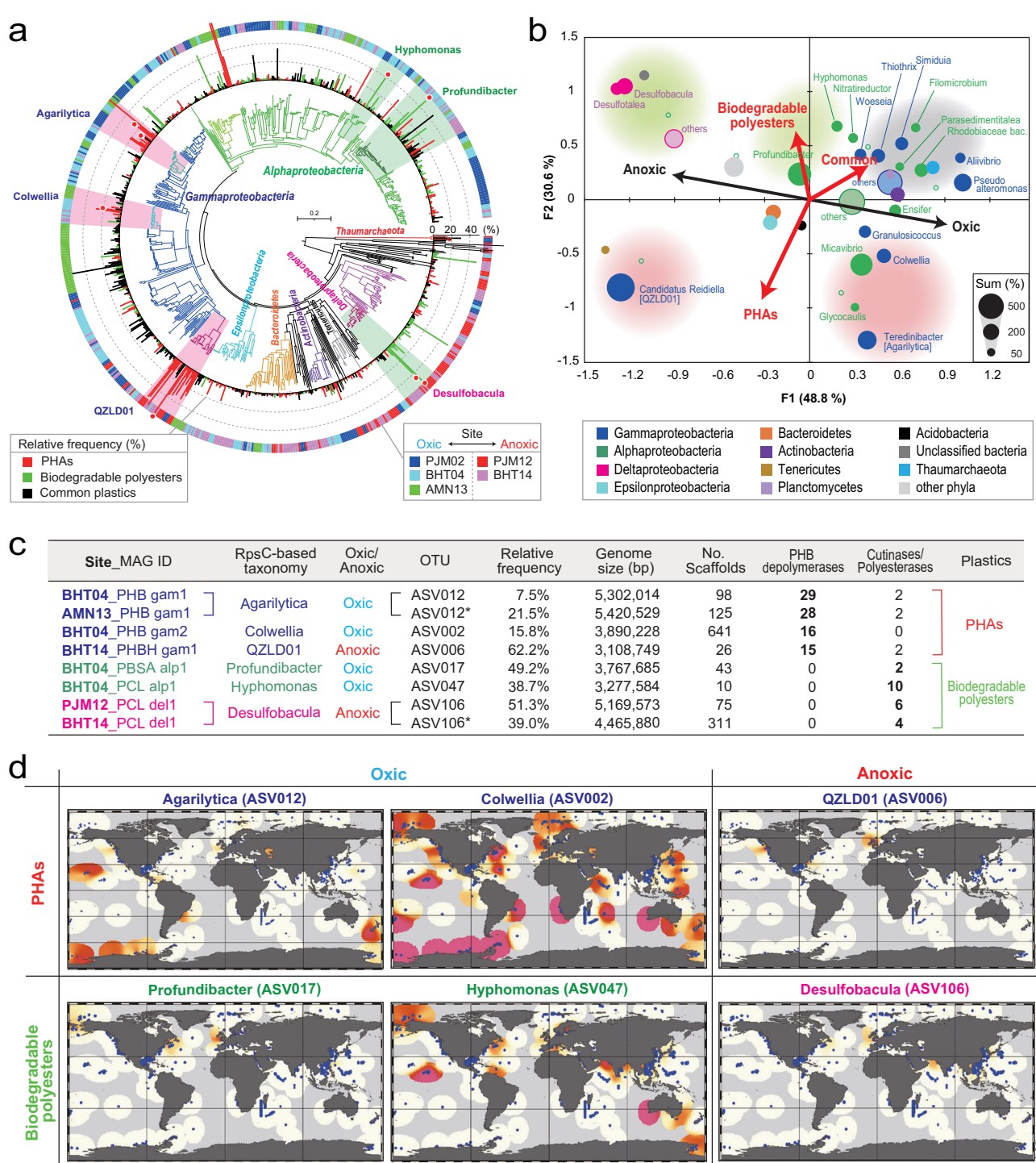

**Fig. 6 | Genomic information and global distribution of dominant microbes within the plastispheres established on PHAs and biodegradable polyesters in the deep-sea. a** Phylogenetic tree of ribosomal protein S3 (RpsC) identified in all metagenomic assemblies. The taxonomy of the proteins is shown by different colors of the nodes. Relative frequency within the communities is shown by a bar chart outside the tree, with different plastic types indicated by color (red, PHAs; green, biodegradable polyesters; black, common plastics). The site and immersion time are shown as colored strips outside of the bar chart. Metagenome-assembled genomes (MAGs) used for further analysis are marked by a red dot, while their taxonomies are also described (Supplementary Data 5–9). **b** Weighted canonical correspondence analysis (wCCA) diagram. The wCCA diagram showing the relationships between five environmental variables (red arrows for the series of the plastic, and black arrows for the redox state), sites (green open circles), and dominant microbes (dots). Dot size indicates the summed relative frequencies among 60 plastispheres (maximum 6000%), while the dot color indicates taxa. For *Alpha-*, *Gamma-*, and *Deltaproteobacteria*, abundant genera (summed relative frequency >50%) were used for the analysis. **c** Statistics for numbers of the MAGs abundantly observed within the plastispheres, which belong to the taxa correlated with PHAs or biodegradable plastics in oxic/anoxic conditions in the wCCA diagram. The number of key genes relate to biodegradation of the plastics is shown. **d** Global distribution of six potential degraders of PHAs and biodegradable polyesters in marine sediment. Full-length 16S rRNA gene amplicon sequences from each MAG were used for the analysis, with publicly available 16S rRNA gene amplicon sequences used as a reference. Source data are provided as a Source Data file.

However, PLLA and non-biodegradable common plastics (PE, PP, PS, PET) did not degrade at all, either onshore or at the deep-sea floor, regardless of depth. Furthermore, the deep-sea bed ecosystem contains a large diversity of aerobic and anaerobic microorganisms that may be able to decompose biodegradable plastics using specific secreted degradation enzymes, and these microorganisms generally have a worldwide distribution. In the future, marine biodegradable plastics that decompose as quickly as possible once they are discharged into the ocean need to be developed. High-performance marine biodegradable plastics that do not biodegrade during use, but instead activate a biodegradation switch when they are discharged into the ocean, ensuring their biodegradation, should also be developed.

## Methods

### Sample preparation

The PHAs, biodegradable polyesters and non-biodegradable common plastics used in this study were purchased from or provided by various companies as listed in Table 2. Polysaccharide triacetates (DS = 3) and polysaccharide acetates (DS = 1.5) were synthesized from a dimethylacetamide/LiCl solution of polysaccharide by adding pyridine and acetic anhydride for different reaction times at 90 °C[27]. Formal names, abbreviations, thermal properties, and chemical structures of all of the samples are summarized in Table 2 and Supplementary Fig. 1.

Injection molding and melt-pressed films were prepared above the melting points of the samples using a HAAKE MiniJet Pro (Thermo Fisher Scientific, Waltham, MA, USA) and Mini Test Press (Toyoseiki, Japan), respectively. The sample sizes for injection molding and film were approximately 1 cm × 3 cm × 0.4 cm, and 4 cm × 4 cm × 300 μm, respectively. In the case of polysaccharides, regenerated gels of neat polysaccharides and polysaccharide acetates (DS = 1.5) were prepared and then pressed at 2 MPa and 100 °C to dry them slowly to obtain films. Films of polysaccharide triacetates were prepared by the solution-casting method from a chloroform solution. The injection-molded samples were put directly into containers divided into nine compartments by Teflon sheets. The containers were sealed with a 1-mm polyethylene mesh to prevent the samples from being released outside but to allow seawater and microorganisms to pass through the containers (Fig. 1c and Supplementary Fig. 2). The melt-pressed films and cast films were sealed with a polyethylene mesh with a 1 mm gap and placed in the container (Supplementary Figs. 2a and 3a). These containers were combined as shown in Supplementary Figs. 2b and 3b and covered with a net as shown in Supplementary Figs. 2c and 3b.

The thickness of the various samples before and after environmental degradation tests was measured, in μm, at five points, using a digital micrometer (Mitsutoyo, Japan) and averaged. For melt-pressed film thickness, five points were measured (at the four corners and in the center of a 4 × 4 cm film), and the average and standard deviation value were obtained. Dimensions were measured in cm to two decimal places using a digital caliper (Shinwa, Japan). Weights were measured in grams to four decimal places using an electronic balance. Weights and dimensional measurements were obtained for 1–2 samples for injection molding and averaged over 2–5 samples for melt-pressed film. Two-dimensional wide-angle X-ray diffraction and small-angle X-ray scattering were performed at the BL03XU beamline of the SPring-8 synchrotron radiation facility to determine crystallinity and crystal structures.

### Sample deployment and recovery

The environmental biodegradation tests were performed on the deep-sea floor at five sites: three bathyal sites [the bathyal seafloor off Misaki port, BMS (35°4.2′N, 139°32.5′E, at a depth of 757 m below sea level); hydrocarbon seepage off Hatsushima Island, BHT (35°0.95′N, 139°13.33′E, at a depth of 855 m below sea level); and a hydrothermal vent off Myojin Knoll, BMJ (32°6.3′N, 139°52.17′E, at a depth of 1292 m below sea level)], and two abyssal sites [the abyssal plain near Kuroshio

Extension Observatory, AKR (32°34.78′N, 143°46.14′E, at a depth of 5503 m below sea level), and the abyssal plain around Minamitorishima Island, AMN (22°59.93 N, 154°24.55E, at a depth of 5552 m below sea level)]. As a reference site that is close to large cities and rivers, the same tests were also performed in a coastal environment [port of JAMSTEC Yokosuka Headquarters, PJM (35°19.18′N, 139°39.04′E, at a depth of 2–6 m)] (Fig. 1a, b and Supplementary Table 1). Figure 1a was made with GeoMapApp (www.geomapapp.org) / CC BY / CC BY (Ryan et al., 2009)[38].

The deployment and recovery of deep-sea samples were conducted by the human-occupied vehicle (HOV) *Shinkai 6500* or remotely operated vehicles (ROVs; *Hyper-Dolphin* and *Kaimei-ROV*), except at AMN, where the free-fall-type deep-sea observatory lander system *Edokko Mark 1* was employed (Supplementary Fig. 4). During the recovery of samples, the sample chambers were placed in a sample box with a lid until the HOV or ROV was recovered on-board, to avoid contamination by seawater from the upper layer. However, during the recovery at AMN, the sample chambers fixed in the lander system went through the water column, leading to potential contamination.

The sample chambers were rapidly dismounted, and the sample specimen was cut into three pieces and subjected to fixation. One piece was placed in sterilized seawater at 4 °C for microbiome studies, one piece was stored at −80 °C for physicochemical characterization and DNA extraction, and one piece was processed for SEM observation. Collected samples were sonicated in distilled water for 30 s to remove surface biofilm. The samples were then vacuum-dried overnight at room temperature. Weights were measured in grams to four decimal places using an electronic balance. Film thickness and dimensions were measured as in the sample preparation section above. Weights and dimensional measurements were of 1–2 sample(s) for injection-molded plastic, and averaged over 2–5 samples for film. Weight loss was determined by Eq. (1):

$$Weight\ loss(wt.\%) = \frac{W_{initial(g)} - W_{final(g)}}{W_{initial(g)}} \times 100 \qquad (1)$$

At BHT, surface sediments near the sample chamber(s) and just below the sample chamber(s) were collected using a push corer (inner diameter = 8.2 cm) manipulated by the HOV *Shinkai 6500*. The recovered sediment core samples were immediately sliced horizontally at 0–1, 2–3, 4–5, and 9–10 cm depths and frozen at −80 °C. Bottom seawater (~2 m above the seafloor) was collected using a Niskin water sampler at both descending target points (~400 m apart) and at the deployment site before landing on the seafloor. The bottom seawater was filtered by Sterivex (PVDF with pore-size of 0.2 μm) on the deck and the filter was immediately frozen at −80 °C.

### Estimated lifetime of plastic bags

Equation (2) was used to calculate biodegradation weight loss per initial area per day (μg/cm²/day), and the data were standardized. This allows for data comparisons of biodegradation tests conducted in different marine environments.

$$V(\mu g/cm^2/day) = \frac{\Delta W(g) \times 10^{-6}}{A_{initial(cm^2)}} \div t(day) \qquad (2)$$

Where $V$ represents the biodegradation rate (μg/cm²/day); $\Delta W$ is the decrease of mass (g); $A$ is the initial surface area (cm²); and $t$ represents time for biodegradation (day).

A typical plastic bag was assumed to be 15 μm thick, 18 cm wide, 42 cm long, and 3 g in weight, as actually measured. Note that the surface area of the film ($S = 3024$ cm²) is twice that of one side, and that the plastic bag is assumed to include two rectangles of 18 cm in width and 42 cm length. From Fig. 2h and Supplementary Fig. 5a–c, the film thickness decreases linearly from the start of decomposition.

Because a film of 15 μm thickness would degrade within 4 months, it is considered that the 15 μm plastic bag would be degraded in aerobic conditions (i.e., before the environment became anaerobic). Hence, the estimated biodegradation time to degradation was calculated using the degradation rate values for PJM02 and BHT03 in aerobic conditions. The residual weight of the plastic bags was calculated using Eq. (3). The fastest and slowest biodegradation rates obtained from experiments conducted with $n = 4$ samples were used.

$$\text{Residual weight(wt.\%)} = 100 - \frac{V(\mu g/cm^2/day) \times S(cm^2) \times t(day)}{3(g)} \times 100 \tag{3}$$

### BOD tests (BOD biodegradability)
Biodegradation in environmental water was evaluated using a BOD measuring device (OxiTop IDS, WTW, Germany). In a cultivation bottle (internal volume 250 mL), 100 mL of seawater was mixed with 100 μL of buffer solution ($Na_2HPO_4 \cdot 2H_2O$ 33.3 g/L, $K_2HPO_4$ 21.8 g/L, $KH_2PO_4$ 8.5 g/L, $NH_4Cl$ 1.7 g/L), 0.5 g/L $NH_4Cl$, 0.1 g/L $Na_2HPO_4$, and 5 mg/L allylthiourea. The sample weight was 6–7 mg. BOD tests were conducted in an incubator (25 °C) for ~1 month, and BOD data were measured daily. BOD biodegradability was calculated from Eqs. (4) and (5), and weight loss was calculated from Eq. (6).

$$\text{BOD} - \text{biodegradability (\%)} = \frac{BOD_s - BOD_b}{ThOD} \times 100 \tag{4}$$

where $BOD_s$ (mg) is the BOD value measured when the sample was added, $BOD_b$ (mg) is the BOD value measured in blank tests, and ThOD is the theoretical oxygen demand (see Eq. 5).

$$\text{ThOD(mg)} = \frac{w(mg)}{M(g/mol)} \times \frac{4x + y - 2z}{4} \times 32(g/mol) \tag{5}$$

where $w$ is the initial sample weight (mg), and $M$ is the molecular weight of the monomer unit ($C_xH_yO_z$) (g/mol).

$$\text{Weight} - \text{loss(\%)} = \frac{W_i - W_f}{W_i} \times 100 \tag{6}$$

where $W_i$ is the initial sample weight (mg) and $W_f$ is the sample weight after microbial degradation (mg).

### SEM
Plastic sample-attached bacteria were fixed with 4% formaldehyde in 0.1 M phosphate buffer containing 150 mM NaCl (pH 7.4), then washed with pure water. The fixed samples were dehydrated with a graded series of ethanol (30%, 50%, 70%, 90%, 100%) for 10 min at room temperature and then substituted twice with *tert*-butyl alcohol. Consecutively, samples were freeze-dried, then sputter coated with gold using an MSP-1S magnetron sputter (Vacuum Device Inc., Japan) before examination in a JCM-7000 (JEOL, Japan) using a secondary electron detector with an acceleration voltage of 5 kV. For 3D-construction of surface erosion images, samples without fixation were washed with pure water using an ultrasonic cleaning bath for several seconds to remove attached bacteria and debris. The cleaned samples were air-dried and sputter-coated with gold. Four SEM images recorded by a backscattered electron detector were reconstructed to a 3D image using SMILEVIEW™ maps software (JEOL, Japan).

### DNA extraction for 16S rRNA gene amplicon sequencing
The cut specimens of collected samples were stored in 40 mL of sterilized Daigo's Artificial Seawater SP for Marine Microalgae Medium (Fujifilm Wako, Osaka, Japan) during transport to the laboratory. Then, following centrifugation at 13,040 × g for 10 min at room temperature, 30 mL of supernatant was discarded, and the tube containing the specimen was shaken at full speed for 2 min using a Vortex-Genie 2 (Scientific Industries) at room temperature. Next, 6 mL of the supernatant was centrifuged at 20,380 × g for 10 min at room temperature, and the pellet was frozen at −20 °C until DNA extraction.

DNA extraction was performed using the Extrap Soil DNA Kit Plus Ver. 2 (BioDynamics Laboratory Inc., Tokyo, Japan) as follows. The pellet was resuspended in 950 μL of extraction buffer and 50 μL of lysis buffer (provided) and transferred to the bead tube (provided). After incubation for 10 min at 65 °C, cells were disrupted by bead-beating for 60 s at 6 m/s using a FastPrep-24 instrument (MP Biomedicals). Bead beating was repeated three times, keeping the sample on ice for 5 min between rounds. Subsequent steps were performed according to the manufacturer's instructions. Purity was estimated based on the absorbance at 260 nm/280 nm (NanoDrop2000c, NanoDrop 2000/2000c ver. 1.5, Thermo Fisher Scientific). The concentration of double-stranded DNA in each sample was measured using a VarioSkan Flash (SkanIt RE for Varioskan Flash ver. 2.4.5, Thermo Fisher Scientific) with Pico Green (Thermo Fisher Scientific).

### 16S rRNA gene amplicon sequencing
A two-step tailed PCR protocol was performed to generate amplicon libraries targeting the V4 hypervariable region of the 16S rRNA gene. The first-round PCR reactions (25 μL, in triplicate) used KOD-Plus Ver. 2 (Toyobo Co., Ltd.) with 240 nM of forward (5′- ACACTCTTTCCCTACACGACGCTCTTCCGATCTNNGTGYCAGCMGCCGCGGTAA-3′)[39] and reverse (5′- GTGACTGGAGTTCAGACGTGTGCTCTTCCGATCTNNGGACTACNVGGGTWTCTAAT −3′)[40] primers and template DNA (<1 ng). The thermal cycling conditions were: 94 °C for 2 min; 25–30 cycles of 98 °C for 10 s, 55 °C for 30 s, and 68 °C for 30 s. Triplicate PCR reactions were then pooled and purified using SPRIselect Beads (Beckman Coulter). Amplicons were eluted with Buffer EB (Qiagen) and quantified by using PicoGreen. To attach dual indexes and sequencing adapters, second-round PCR reactions (50 μL) were performed using KOD-Plus Ver. 2 with 180 nM of forward (5′- CAAGCAGAAGACGGCATACGAGAT[i7]GTGACTGGAGTTCAGACGTGTGCTCTTCCGATCT-3′) and reverse (5′- AATGATACGGCGACCACCGAGATCTACAC[i5]ACACTCTTTCCCTACACGACGCTCTTCCGATCT-3′) primers and 5 ng of the purified first-round PCR product. The thermal cycling conditions were as follows: 94 °C for 2 min; 8 cycles of 98 °C for 10 s, 55 °C for 30 s, and 68 °C for 45 s. Amplicons were then purified using SPRIselect Beads and eluted with Buffer EB. The DNA concentration was quantified by using PicoGreen. Paired-end sequencing (250 bp × 2) was performed using the MiSeq Reagent Kit v2 (500 cycles).

All raw sequences of 16S rRNA gene amplicon sequencing have been deposited at DDBJ (DNA Data Bank of Japan) Sequence Read Archive (DRA) under accession number DRA014821 in BioProject PRJDB14280.

### Read processing for 16S rRNA gene amplicon sequencing
Microbial community analyses were performed using the QIIME2 pipeline version qiime2-2021.4[41]. The raw reads were trimmed using the q2-cutadapt plugin and the trim-paired method with options (--p-front-f GTGYCAGCMGCCGCGGTAA, --p-front-r GGACTACNVGGGTWTCTAAT, --p-error-rate 0.15, and --p-discard-untrimmed). The trimmed reads were clustered and dereplicated into ASVs using the q2-dada2 plugin and the denoise-paired method with options (--p-trunc-len-f 200 and --p-trunc-len-r 190). Taxonomy assignments of the ASVs were performed using QIIME2 with the q2-feature-classifier plugin and classify-sklearn method against the EzBioCloud database[42], and BLASTN (BLASTN ver. 2.12) against the National Center for Biotechnology Information (NCBI) 16S ribosomal RNA database (version 20210814)[43]. Manipulation of the phylogenetic tree of ASVs

was performed using the q2-phylogeny plugin and the align-to-tree-mafft-fasttree pipeline.

## Data analysis for 16S rRNA gene amplicon sequencing

All data were imported into R v4.0.2[44] with qiime2R v0.99.6 (https://github.com/jbisanz/qiime2R/.) and phyloseq v1.34.0[45]. For α- and β-diversity analyses, coverage-based rarefaction was performed using the phyloseq_coverage_rare function in metagMisc v0.0.4 (https://github.com/vmikk/metagMisc.) with an option (coverage = 0.99). For α- and β-diversity analyses, estimate_richness, distance, and ordinate functions in phyloseq were used. ggplot2 v3.3.5[46] and ggrepel v0.9.1 (https://ggrepel.slowkow.com/index.html. https://community.rstudio.com/.) were used for visualization. Phylogenetic trees were drawn using MEGA 11 software ver. 11.0.10[47] after extraction of the top 50 ASVs using the prune_taxa function in phyloseq.

## DNA extraction for metagenome sequencing

Cut specimens, filter-trapped seawater microorganisms, and marine sediments were immediately placed at −80 °C, where they were stored until nucleic acid extraction. DNA was extracted using ZymoBIOMICS DNA/RNA Miniprep Kits (Zymo Research, Irvine, CA, USA) according to the manufacturer's instructions, except for the sediment samples. Briefly, the plastic samples were subjected to bead-beating for 10 min at the maximum speed of the vortex (Vortex-Genie 2, Scientific Industries Inc., Bohemia, NY, USA). DNA from the sediment samples was extracted using an RNeasy PowerSoil Total RNA Kit with DNA Elution Kit (QIAGEN) according to the manufacturer's instructions. The concentration of DNA was measured using a Qubit 3.0 Fluorometer (Thermo Fisher Scientific). The metagenomic sequence libraries were constructed using the Illumina DNA Prep (M) Tagmentation Kit (Illumina, San Diego, CA, USA) with Nextera™ DNA CD Indexes for barcoding (Illumina) according to the manufacturer's instructions. The quality of the constructed DNA libraries was evaluated using an Agilent 2100 Bioanalyzer with a High Sensitivity DNA Kit (Agilent Technologies, Santa Clara, CA, USA) according to the manufacturer's instructions. Paired-end sequencing (150 bp × 2) was performed using the HiSeq X platform at Macrogen Japan. The metagenomic sequences were deposited in the NCBI Sequence Read Archive (SRA) under accession number SRP468711.

## Read processing and estimation of environmental parameters

The demultiplexed raw sequence reads were quality trimmed, and then separately applied to the de novo assembly with scaffolding based on paired-end reads with a $k$-mer of 23 bp, a bubble size of 800 bp, and a minimum scaffold length of 400 bp on the CLC Genomics Workbench version 20.0 (QIAGEN, Venlo, the Netherlands). The raw sequence reads were mapped back to the scaffolds for calculating the coverage of scaffolds using Map Reads and the Contigs function in CLC Genomics Workbench version 20.0 with a length fraction of 0.7 and a similarity fraction of 0.95. The estimated taxonomy of the scaffold was assigned based on the most redundant taxonomic classification of open reading frames (ORFs) called from the scaffold by MetaGeneMark[48] via GhostKOALA ver. 2.0[49], as described elsewhere[50]. G + C content (%) was calculated using the Perl script provided in the MultiMetagenome pipeline[51].

To estimate the redox state and optimal growth rate of the plastisphere that correlated with the environmental conditions, the numbers of each amino acid in the ORFs were calculated for all proteins translated from the ORFs. Then, the existence matrix of 20 amino acids within the whole microbial community was calculated by the sum of the amino acid counts for all ORFs weighted using the coverage of ORFs. The optimal growth temperature of the microbiome ($T_{opt}$) was estimated by the IVYWREL fraction ($F_{IVYWREL}$) using the equation $T_{opt}$ (°C) = $F_{IVYWREL}$ × 937 − 335[52]. To estimate the redox state of the

environment, the average oxidation state of carbon ($Z_C$) of the proteins per $Z_{total}$ was calculated, as described elsewhere[53].

## RpsC-based microbial community analysis

The Kyoto Encyclopedia of Genes and Genomes (KEGG) Automatic Annotation Server (KAAS) ver. 3 April 2015, was used for KEGG orthologous (KO) group assignment with the single-directional best hit method set to 45 as the threshold assignment score[54]. The gene encoding ribosomal protein S3 (rpsC, K02981), a key marker single-copy housekeeping gene for the phylogeny of microorganisms[55], was used for the metagenome-based population analysis[56]. In brief, the taxonomy of RpsC proteins was primarily assigned based on class and genus in the GhostKOALA output, and further curated using a BLASTP search against the RefSeq protein database. The coverage of the scaffold, including the rpsC gene, was multiplied by the length of the RpsC protein, and the ratio of "coverage × length" per summed value of the whole microbial community was calculated as the relative frequency. A weighted multidimensional scale (wMDS) plot was constructed for clustering 60 plastispheres based on the relative frequency matrix of class-level taxa or genus-level taxa for Alphaproteobacteria, Gammaproteobacteria, and Deltaproteobacteria. A wCCA diagram was constructed for correlating the summed relative frequencies of the abundant class/genus and the variables (plastispheres of PHAs, biodegradable polyesters, common plastics; and redox condition, either oxic or anoxic). A principal component analysis (PCA) plot was constructed for clustering BHT plastispheres, BHT deep-sea water microbiomes, and BHT sediments at different depths. The wMDS, wCCA, and PCA were calculated using XLSTAT ECOLOGY (ver. 2020.4.1.1014, Addinsoft, New York, NY, USA).

RpsC proteins with coverage of >10 and a length >150 amino acids were extracted to generate a phylogenetic tree. The proteins were aligned using MUSCLE, and the phylogenetic tree of the RpsC protein sequences was generated using the maximum-likelihood algorithm in the CLC Genomics Workbench 20.0.

## Extraction of metagenome-assembled genomes

Draft genomes (MAGs) were extracted by grouping contigs using a coverage-GC% plot[50]. The contig clusters were then refined by paired-end connections between contig ends[51] that were counted by the Collect Paired Read Statistics function in the CLC Genome Finishing Module (QIAGEN), which allowed for the association of additional scaffolds and repeat regions (for example, rRNA genes), and the removal of incorrect scaffolds[51].

To clarify the MAG quality proposed by the Genomic Standards Consortium (GSC)[57], genome completeness and contamination were analyzed using CheckM ver. 1.1.0[58], while the numbers of tRNAs and rRNAs per MAG were counted using tRNAscan-SE ver. 2.0[59] and RNAmmer ver. 1.2[60], respectively. The identified 16S rRNA sequences coded in the MAGs were associated with the ASVs generated from the amplicon sequences of the 16S rRNA genes within the plastispheres, as described above. For taxonomic assignment of the MAGs, GTDB-tk ver. 3.0 was run with default settings using the GTDB database release 95[61].

The scaffold sequences of each MAG have been deposited at DDBJ/EMBL/GenBank under BioProject PRJNA886482 as biosamples SAMN38046932-SAMN38046948.

## Functional annotation of esterases

Membrane proteins were analyzed by prediction of transmembrane helices using the TMHMM server version 2.0[62]. ORFs encoding esterases were identified by the best BLASTP hit in the ESTHER database ver. May 27, 2020[63], as well as by using an in-house list of depolymerase proteins, including PHA depolymerases, cutinases, petases, PBAT hydrolases, BTA hydrolases, and PBS(A) depolymerases[20] using an e-value cut-off of 1e⁻⁶. Extracellular secretion-type PHA depolymerases were assigned by families Esterase_phb and

Esterase_phb_PHAZ in the ESTHER database at block X, while extracellular polyesterases for degrading biodegradable polyesters were assigned from the in-house depolymerase proteins as well as the families Polyesterase-lipase-cutinase and Carb_B_Bacteria in the ESTHER database at block L.

### Global distribution of dominant microbes

We collected marine sediment 16S rRNA gene metagenome sequencing data from NCBI database, operated under the Illumina platform (HiSeq, MiSeq, NovaSeq, and Genome analyzer), and each data size was >1 MB. The sequencing data were categorized using the same BioProject in the NCBI dataset, and each dataset was denoised and reconstructed under DADA2[64] in QIIME2 v. 2020.82[65] with parameters -p-max-ee 2.0 –p-n-reads-learn 1000,000, as 16S rRNA gene metagenomic assemblies. To collect the distribution patterns of the dominant microbes on PHBs and biopolyesters, their 16S rRNA gene metagenome sequences were identified by BLASTN using rRNA gene metagenomic assemblies as the dataset, with >97% identity and 200-bp matches. The geographic distribution of dominant microbes on PHBs and biopolyesters was reconstructed using Ocean Data View 5.6.03 (https://odv.awi.de/.) with the weighted average of their presence–absence data.

### Reporting summary

Further information on research design is available in the Nature Portfolio Reporting Summary linked to this article.

## Data availability

All the data related to weight loss, thickness reduction, and BOD are provided with this article, Supplementary information and Source data file. All raw sequences of 16S rRNA gene amplicon sequencing have been deposited at DDBJ (DNA Data Bank of Japan) Sequence Read Archive (DRA) under accession number DRA014821 in BioProject PRJDB14280. All raw sequences of metagenomic sequencing have been deposited at NCBI Sequence Read Archive (SRA) under accession number SRP468711 in BioProject PRJNA886482. Detailed information on sequence reads is summarized in Supplementary Table 10. The scaffold sequences of each MAG have been deposited at DDBJ/EMBL/GenBank under BioProject PRJNA886482 as biosamples SAMN38046932 to SAMN38046948. Source data are provided in this paper.

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

## Acknowledgements

We thank all members of the Laboratory of Science of Polymeric Mate-rials, The University of Tokyo, for supporting sample preparation and analyses. Especially we thank Ilangovan Manikandan, Jin Ho Seok, QuiYuan Huang for their assistance in preparing Fig. 4b. We thank the captain and crew of R/Vs *Yokosuka*, *Kairei*, *Shinsei-maru*, and *Kaimei* for their great support of scientific activity during the expeditions YK19-11 (cruise PI: Akinori Yabuki), YK21-08C, YK21-18C, KR20-E01C, KR21-04C, KS-20-1 (cruise PI: Tetsuro Ikuta), KM20-09, and KM21-E02. We extend this thanks to the HOV *Shinkai 6500* team, the ROV *Hyper Dolphin* team, and the ROV *KM-ROV* team. Shiro Matsugaura and Tetsuya Miwa (JAM-STEC) are gratefully acknowledged for providing advice and support during *Edokko Mark 1* deployment. We thank Shino Suzuki, Yuki Sakao, and Tomoka Fukumi for technical assistance in DNA and RNA extraction and sequence library preparation, Yuka Uenaka and Liao Fangyuan for supporting the bioinformatic analyses, and Keiko Tanaka and Sachiko Kawada for supporting the preparation of deployments at JAMSTEC. We thank Rieko Kasaishi, Masataka Furukawa and Chisato Mori for DNA extraction and 16S rRNA gene amplicon sequencing, Erika Usui for supporting the microbial community analysis at NBRC. This paper is based on results obtained from three NEDO projects, Feasibility Study Program "Development of Marine-Biodegradable Plastics" (to T.I., N.I.), the Moonshot Research & Development Program "Research and devel-opment of marine biodegradable plastics with degradation initiation switch function" (JPNP18016) (to T.I., H.N., K-i.Ka.), and "Development of Marine-Biodegradable Assessment Methodology" (PJ-ID 20001845) (to T.I., M.K., A.N., K.Ka) commissioned by the New Energy and Industrial Technology Development Organization (NEDO). A part of this work was supported by the Council for Science, Technology and Innovation (CSTI), Cross-ministerial Strategic Innovation Promotion Program (SIP), "Innovative Technology for Exploration of Deep-sea Resources" (lead

agency: JAMSTEC) (to T.I., S.K., H.M.). This work was supported by a Grant-in-Aid for Scientific Research (A) (grant number: 19H00908 to T.I.) from the Japan Society for the Promotion of Science (JSPS, Japan). The synchrotron radiation experiments were performed at BL03XU, SPring-8, with the approval of the Frontier Softmaterial Beamline (FSBL) (proposal nos: 2019B7270, 2020A7223, and 2021A7222) (to T.I.). We thank Professor Charles Yokoyama, Ph.D., IRCN Executive Manager, The University of Tokyo, for editing a draft of this manuscript.

## Author contributions

T.I. conceived the idea and guided all projects. T.O., K.Ko., M.S., H.M., and A.N. performed sample preparation and biodegradation assessment. S.K. performed SEM experiments. N.I., H.N., R.N., M.T., S.Ka. performed sample deployment and recovery. T.M., M.M., K.H., K.Ka. performed 16S rRNA analysis and data analysis. S.I. carried out metagenome analysis. Y.I. analyzed the global distribution of microbes. T.I., N.I., K-i.Ka, S.K., H.M., A.N., M.K. acquired funds. T.O., N.I., T.M., S.I., M.M., Y.I., S.K., K.Ka, T.I. wrote the paper. All authors discussed the results and commented on the manuscript.

## Competing interests

The authors declare no competing interests.

## Additional information

¹Laboratory of Science of Polymeric Materials, Department of Biomaterial Sciences, Graduate School of Agricultural and Life Sciences, The University of Tokyo, 1-1-1 Yayoi, Bunkyo-ku, Tokyo 113-8657, Japan. ²Research Institute for Marine Resources Utilization, Japan Agency for Marine-Earth Science and Technology (JAMSTEC), 2-15 Natsushima-cho, Yokosuka, Kanagawa 237-0061, Japan. ³Biological Resource Center, National Institute of Technology and Evaluation (NBRC), 2-5-8 Kazusakamatari, Kisarazu, Chiba 292-0818, Japan. ⁴Institute for Extra-cutting-edge Science and Technology Avant-garde Research (X-STAR), Japan Agency for Marine-Earth Science and Technology (JAMSTEC), 2-15 Natsushima-cho, Yokosuka, Kanagawa 237-0061, Japan. ⁵Gunma University Center for Food Science and Wellness (GUCFW), Maebashi, Gunma 371-8510, Japan. ⁶Green Polymer Research Laboratory, Graduate School of Science and Technology, Gunma University, Kiryu, Gunma 376-8515, Japan. ⁷Research Institute for Global Change (RIGC), Japan Agency for Marine-Earth Science and Technology (JAMSTEC), 2-15 Natsushima-cho, Yokosuka, Kanagawa 237-0061, Japan. ⁸Japan BioPlastics Association (JBPA), 5-11 Nihonbashi Hakozaki-cho, Chuo-ku, Tokyo 103-0015, Japan. ⁹Biomedical Research Institute, National Institute of Advanced Industrial Science and Technology (AIST), 1-8-31 Midorigaoka, Ikeda, Osaka 563-8577, Japan. ¹⁰Standardization Promotion Office, National Institute of Advanced Industrial Science and Technology (AIST), 1-1-1 Umezono, Tsukuba, Ibaraki 305-8560, Japan. ✉e-mail: atiwata@g.ecc.u-tokyo.ac.jp

