## [Peer Review File · Nature Communications]

Microbial Decomposition of Biodegradable Plastics on the Deep-sea FloorEditorial Note: Parts of this Peer Review File have been redacted as indicated to maintain the confidentiality of unpublished data and to remove third-party material where no permission to publish could be obtained.

Reviewer #1 (Remarks to the Author):

Omura et al., have investigated degradation of biodegradable plastic in deep sea environments. They have also investigated the microbial community composition and meta genomes of these in biofilms covering the plastic pieces.

I find the study interesting and timely but the analysis remains a bit superficial and some methods are not fully clear. The data are also not well discussed with other data on degradability of biodegradable plastic in the ocean. Below I have provided some more detailed comments orders to sections (there were no line numbers).

Abstract:

- Soil / sediments
- Change 'deep ocean microorganisms with plastic-degrading capacity' to something where the type of plastic is clear.
- Last sentence is detached from study, erase

Intro

- metrics on plastic production and discharge to ocean needs updated referencing (eg PlasticsEurope reports, Mijer et al., 2009); atmospheric transport not mentioned
- fate of plastic in marine realm is not to only stay at the surface but also to beach and degrade (see eg review by Wayman and Niemann, 2001)
- study by Nakajima et al., 2021 is not representative for the entire deep sea – the way it is stated induces the picture that the entire deep sea is covered with ~45000 plastic pieces km⁻². The later statement that "These test sites represented deep-sea floor areas with a high accumulation of plastic waste (BMS, BHT, and AKR), a cold seep (BHT), and a hydrothermal vent (BMJ), and should apply to ocean environments worldwide" is thus also a bit misleading.
- Degradation of biodegradable plastic in the marine realm is not well introduced. Recent papers by eg Lott et al., 2021 or Lopez-Ibanes and Beiras, 2022 should help there.

Biodegradation at deep sea floor

- Not clear how biodegradation is actually measured/defined here

Microbial community analysis

- It's not justifiable to say that "over time only those microorganisms capable of degrading biodegradable plastics remain on the surface". That would mean by definition that all organisms on a plastic surface are eventually plastic degraders.
- What were the reasons for choosing the presented taxonomical level?
- At end of section: the authors try to link microorganisms on their biodegradable plastic to general 'plastisphere' communities based on the fact that they were found on both. That does not make sense as polyoelfins are degraded by other enzymes than polyesters.

Conclusion:

Half of conclusion is more of a 'political statement' (ie which types of plastics should be produced). Though I agree on a personal level with the authors (ie that ideally plastics are designed that can be broken down in a variety of environments), such a call is not a scientific conclusion.

Figs

- Colouration shades are very difficult to determine (eg nmds plots, I can't distinguish many plastic types)

Reviewer #2 (Remarks to the Author):

Omura et al carried out a deep-sea floor incubation of a large number of different plastic materials to determine their biodegradation and microbial colonisation. Authors claim (and emphasise mainly on) the biodegradation of biodegradable plastics and identify the microbes responsible for such biodegradation. Authors show biodegradation only by weight loss, which is a valid method especially for biodegradable plastics, but the methodology used is not clear (see my comments

below). It is also dubious the claim of the microbes involved in such biodegradation without further experimentation (also see my comments below).

Other major comments:

- The finding of biodegradable plastic degradation in deep-sea floors is not surprising, specially of those that are reported here (i.e. PHAs and other polyesters and polysaccharides). It is most interesting that PLLA, as well as other recalcitrant plastics, is not biodegraded, confirming what is known in the literature. The abstract doesn't reflect the data presented, e.g. there is no reference to the common plastic analyses and the breadth of the study.
- The introduction should reflect the global production of these materials worked with and how they distribute within the environment.
- Authors fail to mention a striking finding, i.e. the much lower biodegradation of any material in the deeper seafloor. This could come as a consequence of temperature or pressure. If I understand correctly from the code for the locations (highly confusing all through), those incubated next to the hydrothermal vent (where higher temperatures could have been tested) didn't show higher biodegradation rates, but for some reason, samples must have been much further away from the influence of the vents as the temperatures in this location were similar to others...
- "1/10 for BHT and 1/50 for AMN" what way of reporting biodegradation is this?
- "Evidently, this result is related to the local number and diversity of degrading microorganisms, which will be discussed later" While this may be true, it may also depend on the temperature and pressure.
- No error bars, hence, I assume there was no replication?
- Fig 2e. Is the degradation expected to be lineal over time? This is complex to calculate with the data presented in this study.
- "The number of microorganisms attached to plastic surfaces decreased with sea depth" How was this analysed? Or do authors refer to taxonomic unit numbers?
- "This change might reflect the fact that the sample holders had started to become buried in sediment" This is interesting. It would have been interesting to have a community analysis of the sediment as it is likely to be similar to the plastisphere at this point.
- In order to be able to say the reconstructed MAGs are involved in the biodegradation of X Y and Z, authors should prove these taxonomic species are exclusive to these materials. Could authors clearly show this? Fig 4b is not clear enough. What about a figure such as Extended Data Fig 13?
- The statement from the abstract: "microbiomes identified six new deep ocean microorganisms with plastic-degrading capacity" and other asserting statements that these strains can biodegrade plastics, can only be tested by isolation or SIP experiments... Hence, tone down.
- Instead of using the 16S rRNA to detect the distribution of these 'plastic-degrading' microbes, why not use the depolymerases and esterases?
- Samples were recovered and weighed? What about the biofilm formed? Wasn't it removed? How did authors account for the weight of the biofilm and/or fragmentation?
- "the sample specimen was cut into three pieces and subjected to fixation. One piece was placed in sterilized seawater at 4 °C for microbiome studies, one piece was stored at -80 °C for physicochemical characterization and DNA/RNA extraction, and one piece was processed for SEM observation". This is not clear. What's the difference between the microbiome study and the DNA/RNA extraction? Was transcriptomics even done?

The point-by-point reply to two reviewers are followings.

Reviewer #1:

Omura et al., have investigated degradation of biodegradable plastic in deep sea environments. They have also investigates the microbial community composition and meta genomes of these in biofilms covering the plastic pieces. I find the study interesting and timely but the analysis remains a bit superficial and some methods are not fully clear. The data are also not well discussed with other data on degradability of biodegradable plastic in the ocean. Below I have provided some more detailed comments orders to sections (there were no line numbers).

Reply) Thank you very much for your interest in our research. We have revised whole our paper along with all of your comments. We added the detail methods and additional analyses.

Comments for Abstract)

- Soil / sediments
- Change ‘deep ocean microorganisms with plastic-degrading capacity’ to something where the type of plastic is clear.
- Last sentence is detached from study, erase

Reply) Abstract was revised along the comments (Line 56-58).

Comments for Introduction)

- metrics on plastic production and discharge to ocean needs updated referencing (eg PlasticsEurope reports, Mijer et al., 2009); atmospheric transport not mentioned

Reply) We updated referencing and revised pertinent parts (Ref. 4: Line 66).

- fate of plastic in marine realm is not to only stay at the surface but also to beach and degrade (see eg review by Wayman and Niemann, 2001)

Reply) We updated referencing and revised pertinent parts (Ref. 8: Line 68).

- study by Nakajima et al., 2021 is not representative for the entire deep sea – the way it is stated induces the picture that the entire deep sea is covered with ~45000 plastic pieces km⁻². The later statement that “These test sites represented deep-sea floor areas with a high accumulation of plastic waste (BMS, BHT, and AKR), a cold seep (BHT), and a hydrothermal vent (BMJ), and should apply to ocean environments worldwide” is thus also a bit misleading.

Reply) We updated referencing (Ref.11, 12:Line 68-72). Since there are many different environments in the deep-sea, to avoid misunderstanding, we have modified the text to say only that the experiment was conducted "at five deep-sea bottom locations" near Japan (Line 89-91).

- Degradation of biodegradable plastic in the marine realm is not well introduced. Recent papers by eg Lott et al., 2021 or Lopez-Ibanes and Beiras, 2022 should help there.

Reply) Thank you very much for your suggestion. We add these papers for introduction of BOD degradation and realm degradation. (Ref.16 for Lopez-Ibanes and Beiras: Line 78, Ref.17 for Lott, C. *et al*: Line 78). Furthermore, we added the importance of this research regarding the decomposition of biodegradable plastics at deep-sea floor (Line 83-85).

Comments for Biodegradation at deep sea floor)

- Not clear how biodegradation is actually measured/defined here

Reply) Biodegradation was measured as weight loss in this study. The detail methods describe at the material and methods (Sample deployment and recovery, Line:344~348). Most of samples were not broken physically at deep-sea floors. Accordingly, the degree of weight loss is considered biodegradation in this paper.

Microorganisms covered sample surfaces and degraded from surface to generate the holes. These are direct evidence that biodegradation occurred by the action of microorganisms (Figure 2(f) and Extended data Fig. 6). Furthermore, biodegradable plastics used in this study have already confirmed to have biodegradability in sea waters by laboratory BOD biodegradation tests (Line 177-179, Ref. 15, 18).

Comments for Microbial community analysis

- It's not justify able to say that “over time only those microorganisms capable of degrading biodegradable plastics remain on the surface”. That would mean by definition that all organisms on a plastic surface are eventually plastic degraders.

Reply) This does not mean that only decomposing microorganisms will ultimately remain on the surface. It means that over time, the percentage of decomposing microorganisms will increase (i.e., the diversity will decrease). The wording has been changed to avoid misunderstanding (Line 191~196).

- What were the reasons for choosing the presented taxonomical level?

Reply) If the bar plots were created with a finer taxonomy level, they could not have enough colors and the legends could not fit. Therefore, we have added all taxonomic information to the supporting information section (see Extended Data Figure 7, 9 and supplementary Data 1 and 2).

- At end of section: the authors try to link microorganisms on their biodegradable plastic to general 'plastisphere' communities based on the fact that they were found on both. That does not make sense as polyolefins are degraded by other enzymes than polyesters.

Reply) We are sorry for the confusion. Polysaccharides are hydrophilic materials, but when esterified, they become hydrophobic materials. Therefore, I meant to say that it is similar to the hydrophobic nature of the surface of polyolefins in general. To avoid the misleading, the sentence has been corrected (Line 222-224).

Comments for Conclusion)

Half of conclusion is more of a 'political statement' (ie which types of plastics should be produced). Though I agree on a personal level with the authors (ie that ideally plastics are designed that can be broken down in a variety of environments), such a call is not a scientific conclusion.

Reply) We revised first paragraph and remove the second paragraph along the comments.

Comments for Figs)

- Coloration sheds are very difficult to determine (eg NMDS plots, I can't distinguish many plastic types)

Reply) We are really sorry these colorations, but there are so many samples and testing sites in this paper. Accordingly, we cannot modify these colorations. We added the statistical data sets (X, Y for the NMDS plots) in the Supplementary Data 1 and 2.

Reviewer #2:

Omura et al carried out a deep-sea floor incubation of a large number of different plastic materials to determine their biodegradation and microbial colonization. Authors claim (and emphasize mainly on) the biodegradation of biodegradable plastics and identify the microbes responsible for such biodegradation. Authors show biodegradation only by weight loss, which is a valid method especially for biodegradable plastics, but the methodology used is not clear (see my comments below). It is also dubious the claim of the microbes involved in such biodegradation without further experimentation (also see my comments below).

Reply) Thank you very much for the various useful comments and understanding the concept of our paper. The weight loss method is the only method to directly examine (bio)degradability in deep floors, and subsequently we simultaneously analyzed enrichment of the potential degrading microorganisms on the surface of the plastics. These results clearly showed that the biodegradable plastic could be biodegrading at the deep-sea floor, not by disintegration. For more information, we respond to your comments as below.

Other major comments:

- The finding of biodegradable plastic degradation in deep-sea floors is not surprising, specially of those that are reported here (i.e. PHAs and other polyesters and polysaccharides).

-It is most interesting that PLLA, as well as other recalcitrant plastics, is not biodegraded, confirming what is known in the literature. The abstract doesn't reflect the data presented, e.g. there is no reference to the common plastic analyses and the breadth of the study.

Reply) As for well-known biodegradable plastics such as PHAs and chemosynthetic polyesters in shallow water, soils, and composts, we first elucidated the biodegradation rates in the harsh and cold deep-sea floors. We believe that the rates are essential to develop new biodegradable plastic materials even in deep-sea floor, final resting site of released plastics. Furthermore, we added the importance of this research regarding the decomposition of biodegradable plastics at deep-sea floor (Line 82-84, 98~105).

It was revealed that PLLA and common plastics as PE, PP, PS, PET are not biodegraded in deep-sea floors in this paper. We emphasized them in the text and conclusion (Line 48-50, 120~123).

- The introduction should reflect the global production of these materials worked with and how they distribute within the environment.

Reply) We described the global production at the first paragraph and added the how plastics distribute in environment in Introduction (Line 68-72).

- Authors fail to mention a striking finding, i.e. the much lower biodegradation of any material in the deeper seafloor. This could come as a consequence of temperature or pressure. If I understand correctly from the code for the locations (highly confusing all through), those incubated next to the hydrothermal vent (where higher temperatures could have been tested) didn't show higher biodegradation rates, but for some reason, samples must have been much further away from the influence of the vents as the temperatures in this location were similar to others...

Reply) The deep sea is an extreme environment of low temperature, high water pressure, and low oxygen. Therefore, the biodegradation rate is considered to be slower than that of the coastline. In fact, our experimental results show that the biodegradation rate in the deep sea is extremely slow compared to the shoreline. On the other hand, in our experiments near hydrothermal vents, we believe that there was no difference in biodegradation rates compared to other locations because the seawater is heated and cooled quickly due to the cold water flowing even in the immediate vicinity of the hydrothermal vents. The measured temperature of the BMJ site was 4.6 °C, which was only slightly higher than the other two Bathyal sites (Figure 1b).

- "1/10 for BHT and 1/50 for AMN" what way of reporting biodegradation is this?

Reply) This represents the ratio of the rate of weight loss at the shore and at two different depths (850 m and 5,500 m), which shows that the rate of biodegradation slows down in proportion to depth. We revised to describe more detail clearly as following (Line 127~135) with obtained data.

- "Evidently, this result is related to the local number and diversity of degrading microorganisms, which will be discussed later" While this may be true, it may also depend on the temperature and pressure.

Reply) Thank you very much for the suggestion. Certainly, temperature and pressure might effect on the rate of biodegradability. This speculation was added in the text (Line 136-141). The effects of these factors should be done in future in more.

- No error bars, hence, I assume there was no replication?

Reply) Experiments using films were conducted with n=4-5. The data for films are reliable since they are averages of 4~5 samples (Extended data Fig. 5). Accordingly, the replication of these experiments

seems to be good. On the other hand, the experiment using injection molded products was conducted with $n=1$. This is due to the problem of the capacity of the Shinkai 6500 (manned submersible) to sink at one time.

- Fig 2e. Is the degradation expected to be lineal over time? This is complex to calculate with the data presented in this study.

Reply) Thank you for your comment. As you say, the calculations are complicated because the biodegradation rate is affected not only by crystallinity, surface morphology, etc., but also by environmental conditions such as temperature, type of bacteria present, and oxygen levels. However, based on the data obtained in this study, the fastest and slowest degradation rates at $n=5$ were used to provide a range for the lifetime of the plastic bags as shown in Figure 2e and below.

Although it is not shown in the main results, we have found that in the period of 3~9 months, the conditions are aerobic, and the degradation rate remains almost the same. Therefore, the calculation for Figure 2e was performed using the BHT03 values in Figures 2b and c, using Equation (2) (Line 350~363). However, errors may occur depending on the time required for biofilm formation and anaerobic/aerobic conditions.

- “The number of microorganisms attached to plastic surfaces decreased with sea depth” How was this analyzed? Or do authors refer to taxonomic unit numbers?

Reply) As the reviewer pointed out, it was incorrect to say that the number of microorganisms decreases. Therefore, we have corrected it to say that the diversity of microorganisms has decreased. (Line 191-196)

- “This change might reflect the fact that the sample holders had started to become buried in sediment” This is interesting. It would have been interesting to have a community analysis of the sediment as it is likely to be similar to the plastisphere at this point.

Reply) Thank you very much for your comment. We have metagenomic sequences of the BHT sediment cores of 1, 3, 5, 10 cm below seafloor. The bar chart described below is the RPS3-based community composition of plastispheres and the sediments at site BHT. The bar chart showed that the community structure of plastisphere after 1 year near the sediment got closer to the sediment community especially for “none-degradable” plastics such as PLLA, PP, PS, LDPE. However, the biodegradable plastics such as PHAs, PBSA, PCL showed different types of microbes even in same taxonomic groups such like *Candidatus Reidiella*, Deltaproteobacteria, etc. (data not shown), which suggests that the biodegradable polyesters introduced enrichment of anoxic polyester-degrader as well as the byproduct (monomer) eaters. This result might be interesting, but we consider that it is a bit far

from the main topic of this study; thereby, we will show this result only for answering the reviewers’ comment.

- In order to be able to say the reconstructed MAGs are involved in the biodegradation of X Y and Z, authors should prove these taxonomic species are exclusive to these materials. Could authors clearly show this? Fig 4b is not clear enough. What about a figure such as Extended Data Fig 13?

Reply) Thank you for your suggestion. We updated the additional analyses regarding the 17 highly abundant and added the Extended Figure 15.

- The statement from the abstract: “microbiomes identified six new deep ocean microorganisms with plastic-degrading capacity” and other asserting statements that these strains can biodegrade plastics, can only be tested by isolation or SIP experiments... Hence, tone down.

Reply) As reviewer’s suggestion, we toned down this sentence as “A global survey of deep-sea microbiomes has led to the discovery of six new marine degrading microorganisms that may secrete enzymes that degrade biodegradable plastics.” (Line 56-58).

- Instead of using the 16S rRNA to detect the distribution of these ‘plastic-degrading’ microbes, why not use the depolymerases and esterases?

Reply) Thank you very much for this suggestion. At this moment, information of the 16S rRNA sequences is extremely more than the sequences of enzymes from the public metagenomic projects. We will consider the reviewer's suggestion when more public resources will be available in the future.

- Samples were recovered and weighed? What about the biofilm formed? Wasn’t it removed? How did authors account for the weight of the biofilm and/or fragmentation?

Reply) Biofilms on the sample surfaces were removed by ultrasonic cleaning, vacuum dried, and weighed. Basically, thick films and injection-molded products were used in the experiment, so ultrasonic cleaning did not cause them to fragmentate (Line 344-348).

- “the sample specimen was cut into three pieces and subjected to fixation. One piece was placed in sterilized seawater at 4 °C for microbiome studies, one piece was stored at -80 °C for physicochemical characterization and DNA/RNA extraction, and one piece was processed for SEM observation”. This is not clear. What’s the difference between the microbiome study and the DNA/RNA extraction? Was transcriptomics even done?

Reply) The institutions that analyzed the microbiome analysis and the metagenome analysis are different. In addition, because microorganisms were isolated at the same time as the microbiome

analysis, sample storage temperatures and extraction methods are different. To avoid the confusion, we deleted the statement about the RNA (DNA extraction and metagenome sequencing : Line 446-459). We hope that we will perform the metatranscriptomics analysis in near future.

Reviewer #1 (Remarks to the Author):

the MS has improved and many comments initially raised are clarified. What is still not clear is the degradation testing and which controls were made to verify that the reduction in mass/thickness was not the result of sample treatment. More comments below:

Document must be checked for language and grammar. I'm only mentioning here one example from L 40/41: "their [biodegradable plastic] fate on the deep sea floor, the final resting site marine pollutants and 65% of earth's surface is unknown" implies that 65% of Earth's surface is finally resting there and not that 65% of the Earth surface are deep sea floor. Similarly L43: "robust microbial decomposition" is odd wording. Do the authors mean that they have robust results for decomposition, that this is a wide spread process? – quite unclear to me.

Degradation test is still unclear to me. The authors measured reduction in thickness, how did they account for abiotic degradation (ie what was the control)? Also, how did the authors account for material loss due to sample treatment procedure (again, what was the control)?

L48, PLLA not defined

L 90, unclear why BMS, BHT, and AKR are high accumulation sites, give reference

L 133, unclear sentence. Why shouldn't there be PHA degradation at a hydrothermal vent site?

And what is the explanation with the heating/cooling thereafter? Is that the reason why the PHA was degraded?

L 142 this paragraph about polysaccharides comes out of the blue, unclear why they are introduced at this position

L153 So the bioplastics such as PAH show a decreasing degradation trend in biodegradation with depth and this one an increasing?

L194 ff That is not true. In nature there will always be a 'zoo' of organisms in a biofilm, some of these, possibly the majority will use the substrate but definitely not all.

L 200 did you measure O₂? Seems very unlikely that the bottom waters at the deep sea floor are low in O₂

Reviewer #2 (Remarks to the Author):

Omura et al carried out a deep-sea floor incubation of a large number of different plastic materials to determine their biodegradation and microbial colonization. Authors claim (and emphasize mainly on) the biodegradation of biodegradable plastics and identify the microbes responsible for such biodegradation. Authors show biodegradation only by weight loss, which is a valid method especially for biodegradable plastics, but the methodology used is not clear (see my comments below). It is also dubious the claim of the microbes involved in such biodegradation without further experimentation (also see my comments below).

Reply) Thank you very much for the various useful comments and understanding the concept of our paper. The weight loss method is the only method to directly examine (bio)degradability in deep floors, and subsequently we simultaneously analyzed enrichment of the potential degrading microorganisms on the surface of the plastics. These results clearly showed that the biodegradable plastic could be biodegrading at the deep-sea floor, not by disintegration. For more information, we respond to your comments as below.

Re-reply: Weight loss is not the only method, specially when the intention is to determine WHO is degrading the material. The measurements and OMICs techniques used are only a prediction. Reflect in the text.

- The introduction should reflect the global production of these materials worked with and how they distribute within the environment.

Reply) We described the global production at the first paragraph and added the how plastics distribute in environment in Introduction (Line 68-72).

Re-reply: I didn't find the global production of biodegradable plastics. For example, which is the global production of PHAs? It would be interesting to know. Probably very low....

- Fig 2e. Is the degradation expected to be lineal over time? This is complex to calculate with the data presented in this study.

Reply) Thank you for your comment. As you say, the calculations are complicated because the biodegradation rate is affected not only by crystallinity, surface morphology, etc., but also by environmental conditions such as temperature, type of bacteria present, and oxygen levels. However, based on the data obtained in this study, the fastest and slowest degradation rates at n=5 were used to provide a range for the lifetime of the plastic bags as shown in Figure 2e and below. Although it is not shown in the main results, we have found that in the period of 3~9 months, the conditions are aerobic, and the degradation rate remains almost the same. Therefore, the calculation for Figure 2e was performed using the BHT03 values in Figures 2b and c, using Equation (2) (Line 350~363). However, errors may occur depending on the time required for biofilm formation and anaerobic/aerobic conditions.

Re-reply: I truly don't think biodegradation will be lineal due to community successions, aerobic to anaerobic transitions, etc.

- "This change might reflect the fact that the sample holders had started to become buried in sediment" This is interesting. It would have been interesting to have a community analysis of the sediment as it is likely to be similar to the plastisphere at this point.

Reply) Thank you very much for your comment. We have metagenomic sequences of the BHT sediment cores of 1, 3, 5, 10 cm below seafloor. The bar chart described below is the RPS3-based community composition of plastispheres and the sediments at site BHT. The bar chart showed that the community structure of plastisphere after 1 year near the sediment got closer to the sediment community especially for "none-degradable" plastics such as PLLA, PP, PS, LDPE. However, the biodegradable plastics such as PHAs, PBSA, PCL showed different types of microbes even in same taxonomic groups such like *Candidatus Reidiella*, *Deltaproteobacteria*, etc. (data not shown), which suggests that the biodegradable polyesters introduced enrichment of anoxic polyester-degrader as well as the byproduct (monomer) eaters. This result might be interesting, but we consider that it is a bit far from the main topic of this study; thereby, we will show this result only for answering the reviewers' comment.

Re-reply: Sediment communities would greatly improve the study. Please include them in the text and not only in the reply to reviewers. Even a PCA or NMDS plot would be fantastic!

10 September 2023

Point-by-Point Reply to Reviewers #1 and #2

RE: Manuscript no. NCOMMS-22-51041-T

Title: Microbial Decomposition of Biodegradable Plastics on the Deep-sea Floor

Thank you very much for your comments to our revised paper “Microbial Decomposition of Biodegradable Plastics on the Deep-sea Floor” submitted to *Nature Communications* (MS no. NCOMMS-22-51041A-Z). We have revised our manuscript by adding new figures and experimental results that we believe will help to clearly understand the experimental methods and results.

We understand that you are unclear whether the plastic degradation that we observed at the deep-sea floor is physical deformation or microbial degradation and you are eager to clarify this point. Therefore, we present the results of several control experiments that prove that this is microbial degradation and not physical deformation. We have also created a new figure that clearly shows that the samples recovered from the deep-sea have not undergone physical deformation by the ultrasonication used to wash the samples, but have been degraded by microorganisms. These are explained in detail below.

In this revised paper, new figures have been added to enhance the understanding of the sample preparation, the deep-sea experiments, and the progression of microbial attachment and decomposition on the actual samples recovered from the deep sea. The additional Figures and Tables are explained first, followed by responses to comments.

<List of additional New Figures and Tables>

Main Figure 1c-g: Pictures of sample preparation used in this study, sample deployment in the deep-sea floor and recovery of sediments under the samples are added.

Main Figure 2: Biodegradation of PHBH injection-molded and melt-pressed film samples placed at the deep-sea floor and models of microbial degradation

Main Figure 3: Analysis of PBSA, PLLA, and PP melt-pressed films placed at deep-sea floor

Main Figure 4b: The data of Biodegradation rate of samples placed at the deep sea off Hatsushima Island for 8 months (BHT08) was added.

Main Figure 4c: Plot of the biodegradation rate of each plastic against depth, showing that the rate slows with depth.

Extended Data Tables 3-6: Data of samples placed at PJM02, BHT03, BHT08, BMS05.

Extended Data Figure 1a: Categories, names and abbreviations of all samples used in this study

Extended Data Figure 5: Analysis of films other than PHBH, PLLA, PBSA, and PP after recovery from deep sea

Extended Data Figures 9-11: Microbial community analysis of deep-sea sediments off Hatsushima Island

Reply to Reviewer #1

Comment) the MS has improved and many comments initially raised are clarified. What is still not clear is the degradation testing and which controls were made to verify that the reduction in mass/thickness was not the result of sample treatment. More comments below:

Reply) To prove that the plastics in this experiment were not subject to physical deformation at the deep-sea floor, but underwent microbial biodegradation, we present three sets of experimental results:

- (1) We newly analyzed film samples placed off Hatsushima Island for 8 months (BHT08) to analyze changes in deep-sea degradation over time. The new figures were combined with the results of the previous three months. New figures (Fig. 1, Fig. 2, Fig. 3, and Extended Data Fig. 5) for visualization of shape, biofilm formation, and surface degradation of samples recovered from the deep-sea. We also report that the degradation is not only weight loss, but also decreased sample thickness and changed surface morphology.
- (2) A control experiment simulating a deep-sea environment (water pressure of 10 MPa at 4°C using sterile seawater in the laboratory for 4 months).
- (3) Results of control experiments using microbeads, showing that the progress of deep-sea degradation is the same as that of enzymatic degradation in the laboratory.

(1) Visualization of shape, biofilm formation, and surface degradation of samples recovered from the deep-sea

In new Figures 1c and 1d, we provide a visual representation of how the samples were contained when submerged on the deep-sea floor. The injection molded samples were placed in a custom-made sample holder divided by Teflon sheets, with mesh on the top and bottom to prevent the sample from falling out. Then, two of these sets were placed inside a PET bottle with holes. When the test samples were films, the entire sample was placed individually in a mesh bag. These were collected and placed in PET bottles with holes. Four PET bottles containing the samples were then connected together, wrapped in tennis nets, and placed on the deep-sea floor. Therefore, we do not believe that the samples will physically degrade due to ocean currents or other factors.

New Main Figure 1c-g | Sample deployment at deep-sea floor.

Next, the shape and SEM surface observations of injection-molded samples recovered from the deep-sea are shown in new Figure 2a-d. As representative examples, we show the overall shape and surface SEM images of the recovered PHBH injection molded samples from the shore (PJM12), off Hatsushima Island (BHT14), and off Minamitorishima Island (AMN13) after approximately one year (See Main Table 1 for abbreviations, etc.). The injection molded sample is a solid and hard sample, and thus no physical deformation occurred. Degradation progressed mainly in the direction of the sample thickness, especially in the shore samples, which were reduced by 700 μm (Figure 2c). The decrease was 110 μm for the samples placed off Hatsushima Island (BHT14), and 10 μm for those off Minamitorishima Island (AMN13). Surface observations by SEM show a change from a smooth surface to a bumpy surface in these samples (Figure 2d). This series of results suggests that the samples were degraded from the surface in the deep-sea by microbial degradation, rather than by physical deformation.

New Main Figure 2a-d | Biodegradation of PHBH injection-molded samples placed at the deep-sea floor.

New Main Figure 2e–h shows the shape (2e), stereomicroscopic (2f) and SEM images (2g) of a PHBH film sample (BHT08) recovered from the deep-sea, as well as the weight loss and thickness reduction (2h). The film shape indicates that no physical deformation occurred. The surface of BHT08 is greenish and slimy, indicating that microorganisms have attached and formed a biofilm. SEM images (2g, center) also show that many microorganisms are adhering to the surface. In the photograph after ultrasonic cleaning, the film has not physically deformed and has retained its shape (2e, BHT08 after washing). However, stereomicroscopic observation reveals small holes in the film (2h). SEM photographs also show that the surface is uneven and degradation has occurred. Furthermore, as shown in Figure 2h, a decrease in thickness as well as weight was observed. On the basis of these results, the film degradation is considered to be microbial degradation, rather than physical deformation.

New Main Figure 2e–h | Biodegradation of PHBH melt-pressed film samples placed at the deep-sea floor.

The surface morphology of PHBH film placed at the deep-sea floor off Hatsushima Island (BHT) for 8 months (Above New Main Fig.2g) was compared to that of PHB film degraded by microorganisms isolated from soil reported by Nishida and Tokiwa (1992) (below Extra Figure (b)). Comparing the two SEM photographs, both show exactly the same degradation manner, with spherical holes and unevenness surfaces. This result also suggests that film degradation in the deep-sea floor was degraded by deep-sea microorganisms, similar to film degradation by soil microorganisms.

Extra Figure: (a) SEM observation of PHBH film surface placed at the deep-sea floor off Hatsushima Island (BHT) for 8 months in this study. (b) SEM observation of PHB film surface degraded by microorganisms isolated from soil reported by Nishida and Tokiwa, J. Appl. Polym. Sci. 46, 1467-1476 (1992)²⁴.

New Main Figure 3a–c shows the changes in film morphology, surface SEM images, residual weight, and film thickness for poly(butylene succinate-*co*-adipate) (PBSA) and poly(L-lactic acid) (PLLA) as biodegradable polyesters, and polypropylene (PP) as non-biodegradable common plastic, respectively, which were placed off Hatsushima for 3 and 8 months. SEM observation of film surfaces of other biodegradable polyesters (PHB, PHBV, P3HB4HB, PBS, PCL, PBAT) and non-biodegradable common plastics (PE, PS, PET) are shown in Extended Data Figure 5.

The surface SEM image of PBSA also showed the formation of biofilm on the surface (Main Fig. 3a). In the SEM image of the surface, unevenness and holes were observed, indicating that the surface had been decomposed by microorganisms. Therefore, it is considered that microbial degradation of PBSA in the deep sea progresses in the same manner as that of PHBH (Main Fig. 2e). However, the weight loss rate and changes in film thickness were much slower/lower than those for PHBH.

In the case of PLLA, microorganisms were observed adhering to the film surface, but after 8 months, the film surface was very smooth, indicating that no degradation had occurred, despite that PLLA is categorized in biodegradable polyesters. The sample weight and thickness remained unchanged during submersion at the deep-sea floor (Main Fig. 3b). This is thought to be due to the absence of bacteria capable of degrading PLLA among the attached microorganisms. PLLA clearly does not degrade at the deep-sea floor as well as it does not in soil and rivers.

In the case of PP, which is a representative non-biodegradable common plastic, it was observed that the amount of microorganisms attached to film surface was low (Main Fig. 3c). This may be because the surface of PP is hydrophobic, unlike biodegradable polyesters, which have hydrophilic ester groups. It was clear that the PP did not degrade at the deep-sea floor.

In other words, the results for the PLLA and PP films, which did not change at all, can be interpreted as a control experiment.

All of these samples were measured after ultrasonic washing, providing evidence that physical deformation did not occur during the experimental processing.

New Main Figure 3 | Analysis of PBSA, PLLA, and PP melt-pressed film samples placed at the deep-sea floor.

(2) Results of a control experiment simulating a deep-sea environment (water pressure 10 MPa, 4°C, using sterile seawater in the laboratory)

To prove that the degradation observed in the deep-sea is microbial degradation, as a control experiment, a 4-month degradation test was conducted using sterile seawater at 4°C and about 10 MPa (equivalent to about 1000 m depth) in a laboratory setting. Sterile seawater was made from seawater collected from the deep sea off Hatsushima Island. In the figure below,

- (a) shows the experimental apparatus, which can be subjected to high water pressure. Simulating the deep-sea off Hatsushima Island (BHT, depth 855 meters), the temperature was set to 4°C, water pressure of about 10 MPa was applied. PHBH and PP films were placed in the apparatus for 4 months.
- (b) shows the weight retention of the PHBH films in sterile seawater at 10 MPa or at deep-sea floor off Hatsushima Island (BHT). The weight of film in the sterile seawater at 10 MPa did not change, despite that the weight of film at deep-sea decreased.
- (c) Shows the weight retention of the PP films in sterile seawater at 10 MPa or at deep-sea floor off Hatsushima Island (BHT). In the case of PP, both films did not change at all.

Therefore, we can say that the observed degradation in the deep-sea was due to microbial action.

[redacted]

(3) Results of control experiments using microbeads, showing that the progress of deep-sea degradation is the same as enzymatic degradation.

We believe that the degradation observed in the samples recovered from the deep-sea floor was not caused by physical deformation or hydrolysis, but by enzymes secreted by microorganisms. As a control experiment to provide evidence for this, we present the results of the following experiment using P(3HB) microbeads. In the figure below,

(a) shows the P(3HB) microbeads used in the experiment. The beads are spherical (diameter about 50 to 100 µm) with very smooth surfaces.

(b) shows the morphology of P(3HB) microbeads after immersion in 0.1 M phosphate buffer (pH 7.4) for 30 days. The surface morphology of the P(3HB) microbeads is the same as before immersion. This means that simply immersing in the buffer does not cause any change in morphology.

(c) shows the results of an enzymatic degradation test using extracellular PHB depolymerase from *Ralstonia pickettii* T1, which was isolated and purified by our team, in 0.1 M phosphate buffer (pH 7.4) for 7 days. The enzyme eroded into the interior of the microbeads, leaving the overall shape intact, but progressively creating holes of different sizes. This indicates that the depolymerases are reaching the inside of the microbeads while degrading them. This enzymatic degradation is a common phenomenon not only in microbeads but also in films and fibers.

(d) is an SEM picture of a microbead placed on the deep-sea floor off Hatsushima (855 m depth) for 3 months. As with the enzymatic degradation in the laboratory, the degradation penetrated into the interior of the microbead, leaving the overall shape but opening holes. This is an evidence that the degradation at the deep-sea floor is caused by depolymerases secreted by microorganisms, not by physical deformation.

Furthermore, the molecular weight of the microbeads was measured before and after immersion at the deep-sea floor. There was no change in the molecular weight of the polymer. Therefore, hydrolysis by seawater and high hydrostatic pressure did not occur.

These experiments are not described in the revised manuscript. If the referees and editor think that is appropriate, we can include this result into manuscript, since this result is original.

[redacted]

Collectively, from our results, we conclude that the degradation of biodegradable plastics in the deep-sea was caused by microbial biodegradation, not hydrolysis or physical deformation.

Comment) Document must be checked for language and grammar. I'm only mentioning here one example from L 40/41: "their [biodegradable plastic] fate on the deep sea floor, the final resting site marine pollutants and 65% of earth's surface is unknown" implies that 65% of Earth's surface is finally resting there and not that 65% of the Earth surface are deep sea floor. Similarly L43: "robust microbial decomposition" is odd wording. Do the authors mean that they have robust results for decomposition, that this is a wide spread process? – quite unclear to me.

Reply) We sincerely apologize for the unclear meaning of the text. Our revised manuscript has been sent for English language editing throughout by a native English-speaking scientist.

[redacted]

Comment) Degradation test is still unclear to me. The authors measured reduction in thickness, how did they account for abiotic degradation (ie what was the control)? Also, how did the authors account for material loss due to sample treatment procedure (again, what was the control)?

Reply) We explained our detailed response in the reply to first comment.

Comment) L48, PLLA not defined

Reply) L48, We have defined PLLA [Poly(L-lactic acid)].

Comment) L 90, unclear why BMS, BHT, and AKR are high accumulation sites, give reference.

Reply) The amount of plastic waste at each location has been measured, as shown below, but these data have not yet been published and there is no comparison with other locations, and thus this part of the paper has been deleted.

Area	Plastic density (items/km ²)	Reference
AKR	4561	Nakajima et al. (2021)
BHT	860	Nakajima et al. (in prep)
BMS	6230	Nakajima et al. (in prep)

Comment) L 133, unclear sentence. Why shouldn't there be PHA degradation at a hydrothermal vent site? And what is the explanation with the heating/cooling thereafter? Is that the reason why the PHA was degraded?

Reply) At a hydrothermal vent site (i.e., BMJ), PHB was degraded as shown in Main Figure 4b. We did not submerge other PHA samples (PHBV, PHBH, P3HB4HB) at the hydrothermal vent site. Accordingly, we have no data for these plastics.

Although the temperature is indeed high at the hydrothermal vent, the water is immediately cooled by the surrounding 4°C seawater, resulting in a water temperature of 4.6°C at the location where the sample was placed (Detail information test site of deep-sea lists in Main Table 1). Therefore, PHB films were degraded at the same degradation rate as in other deep-sea sites with similar depths [off Hatsushima (BHT) and off Misaki (BMS)] (as explained in L241-L244).

Comment) L 142 this paragraph about polysaccharides comes out of the blue, unclear why they are introduced at this position.

Reply) In the revised manuscript, we have separated this section from the section on biodegradable polyesters (L266-L287).

Comment) L153 So the bioplastics such as PHA show a decreasing degradation trend in biodegradation with depth and this one an increasing?

Reply) On the basis of our results, the rate of degradation in the deep-sea decreases as the depth increases as shown in Main Figure 4c. We have made this point clearly in the revised text (L244-L248).

Comment) L194 ff That is not true. In nature there will always be a 'zoo' of organisms in a biofilm, some of these, possibly the majority will use the substrate but definitely not all.

Reply) Thank you very much for this comment. I agree with your opinion that a large number of microorganisms exist in biofilm. We revised text as followings;

L294-L302: The diversity indexes of microorganisms present on biodegradable polyesters decreased with sea depth, and also decreased while biodegradation rates increased (Fig. 5a, b). A very large number of microorganisms exist in nature. Therefore, in the initial stage, many

types of microorganism are expected to adhere to the surface of the plastic. Among them, only a few microorganisms are capable of degrading biodegradable plastic. Over time, microorganisms that can degrade plastic are able to multiply, resulting in a relative increase in the percentage of degrading microorganisms^{32,33}. In fact, as shown in Fig. 5a, b, comparing, for example, BHT04 with BHT14, we can see that the microbial diversity decreased over time (i.e., with the progress of degradation).

Comment) L 200 did you measure O₂? Seems very unlikely that the bottom waters at the deep sea floor are low in O₂

Reply) Indeed, the O₂ concentration of seawater decreases down to 1000 m, but after that it rises again, and thus this statement was deleted.

Reply to Reviewer #2

1st Comment) Omura et al carried out a deep-sea floor incubation of a large number of different plastic materials to determine their biodegradation and microbial colonization. Authors claim (and emphasize mainly on) the biodegradation of biodegradable plastics and identify the microbes responsible for such biodegradation. Authors show biodegradation only by weight loss, which is a valid method especially for biodegradable plastics, but the methodology used is not clear (see my comments below). It is also dubious the claim of the microbes involved in such biodegradation without further experimentation (also see my comments below).

1st Reply) Thank you very much for the various useful comments and understanding the concept of our paper. The weight loss method is the only method to directly examine (bio)degradability in deep floors, and subsequently we simultaneously analyzed enrichment of the potential degrading microorganisms on the surface of the plastics. These results clearly showed that the biodegradable plastic could be biodegrading at the deep-sea floor, not by disintegration. For more information, we respond to your comments as below.

2nd Comment, Re-Reply) Weight loss is not the only method, specially when the intention is to determine WHO is degrading the material. The measurements and OMICs techniques used are only a prediction. Reflect in the text.

2nd Reply) Thank you very much for your comment. As you mentioned, weight-loss measurement and OMICs techniques used in this study are not the direct evidence that microorganisms degrade biodegradable plastics. Accordingly, we added several new figures and explanation in main text and showed additional experimental results as shown above in this reply to reviewers. And also, along your comment, we added the following sentence in the text of Line 378-390.

L378-L390 in text added as follows

To clarify these matters, it will be necessary in the future to isolate degrading microorganisms that appear to be capable of degrading certain biodegradable plastics and to isolate the degrading enzymes they secrete.

1st Comment) The introduction should reflect the global production of these materials worked with and how they distribute within the environment.

1st Reply) We described the global production at the first paragraph and added the how plastics distribute in environment in Introduction (Line 68-72).

2nd Comment, Re-Reply) I didn't find the global production of biodegradable plastics. For example, which is the global production of PHAs? It would be interesting to know. Probably very low....

2nd Reply) The current production of biodegradable plastics is added in the text (L79). It is still very small, but we expect that production will increase more and more in the future.

Biodegradable plastics were produced approximately 1.14 million tons in 2022. PHA was 25,000 tons.

1st Comment) Fig 2e. Is the degradation expected to be lineal over time? This is complex to calculate with the data presented in this study.

1st Reply) Thank you for your comment. As you say, the calculations are complicated because the biodegradation rate is affected not only by crystallinity, surface morphology, etc., but also by environmental conditions such as temperature, type of bacteria present, and oxygen levels. However, based on the data obtained in this study, the fastest and slowest degradation rates at n=5 were used to provide a range for the lifetime of the plastic bags as shown in Figure 2e and below. Although it is not shown in the main results, we have found that in the period of 3~9 months, the conditions are aerobic, and the degradation rate remains almost the same. Therefore, the calculation for Figure 2e was performed using the BHT03 values in Figures 2b and c, using Equation (2) (Line 350~363). However, errors may occur depending on the time required for biofilm formation and anaerobic/aerobic conditions.

2nd Comment, Re-Reply) I truly don't think biodegradation will be lineal due to community successions, aerobic to anaerobic transitions, etc.

2nd Reply) As the Reviewer says, the degradation rate obviously slows down when the environment changes from aerobic to anaerobic due to the formation of biofilm and coverage of the plastic by seabed soil. Therefore, as you point out, the degradation will not be linear. Our explanation about this was unclear, and we have clearly corrected this point in the revised manuscript.

What we want to show in this paper is a calculation of how long a plastic bag, 15- μ m thick, made of biodegradable polyester, would take to degrade in the deep-sea. In order not to mislead, we have limited the calculations to PHAs. As shown in Main Figure 2h, the thickness of PHBH at the deep-sea bottom off Hatsushima Island decreased by about 30 μ m in 3 months. The conditions are aerobic throughout this 3-month period confirmed by analysis of microbial community on the film surface of samples as shown in Figure 5c, BHT04. Thus, degradation of a 15- μ m film will occur entirely in aerobic conditions. The degradation of the PHA bag was calculated to occur in about 18-58 days in deep water off Hatsushima Island (BHT), as shown in Main Figure 4d.

L252-L261 in text is revised as follows

Using the degradation rates of biodegradable plastics at the deep-sea floor obtained in this study, we calculated the time to complete biodegradation of plastic bags at the shore and in the deep-sea off Hatsushima (BHT), assuming that plastic bags of 15- μ m thickness were made of four different PHAs. From the experimental results, PHA degrades on average by about 35 μ m of thickness in 3 months, and thus 15 μ m of degradation can occur well within the experimental period. At the shore, the calculated degradation time was 6 days for PHBH, which was considered to be the fastest degradation, and 13 days for PHB, which was the slowest degradation (Fig. 4d). These values are almost the same as those previously reported²⁵. However, at the deep-sea floor, the biodegradation periods were calculated to be about 19 days for PHBH and 58 days for PHB (Fig. 4d).

1st Comment) “This change might reflect the fact that the sample holders had started to become buried in sediment” This is interesting. It would have been interesting to have a community analysis of the sediment as it is likely to be similar to the plastisphere at this point.

1st Reply) Thank you very much for your comment. We have metagenomic sequences of the BHT sediment cores of 1, 3, 5, 10 cm below seafloor. The bar chart described below is the RPS3-based community composition of plastispheres and the sediments at site BHT. The bar chart showed that the community structure of plastisphere after 1 year near the sediment got closer to the sediment community especially for “none-degradable” plastics such as PLLA, PP, PS, LDPE. However, the biodegradable plastics such as PHAs, PBSA, PCL showed different types of microbes even in same taxonomic groups such like *Candidatus Reidiella*, Deltaproteobacteria, etc. (data not shown), which suggests that the biodegradable polyesters introduced enrichment of anoxic polyester-degrader as well as the byproduct (monomer) eaters. This result might be interesting, but we consider that it is a bit far from the main topic of this study; thereby, we will show this result only for answering the reviewers’ comment.

2nd Comments, Re-Reply) Sediment communities would greatly improve the study. Please include them in the text and not only in the reply to reviewers. Even a PCA or NMDS plot would be fantastic!

2nd Reply) In accordance with your suggestion, we have analyzed the sedimentary assemblage. We have also conducted metagenomic analysis of plastispheres in the deep-sea water near and surrounding the sample chamber. We have included the results in the main text, as follows. We have also added the necessary figures as support (Extended Data Figs. 8–10). Thank you very much for your suggestion.

L315-L325 in text is added as follows

Metagenomic analysis of plastispheres in the deep-sea water near and surrounding the sample chamber at BHT, as well as in the sediments below the seafloor (1, 3, 5, and 10 cm under the chamber), was also performed. The results obtained were compared with the microbial community attached to the surface of biodegradable plastic immersed in BHT for 4 and 14 months (Extended Data Fig. 8). The microbial population attached to the plastic surface after 14 months became closer to the microbial community in the sediment (1–5 cm) (Extended Data Figs. 9 and 10). These results indicate that long-term immersion may have affected the plastispheres via the sediment covering the sample chambers and that anaerobic microorganisms such as Deltaproteobacteria, “*Candidatus Reidiella*,” and others may have attached from the sediment and grown in the plastispheres (Extended Data Fig. 11).

Extended Data Fig. 10 | Metagenome-based microbial community compositions of plastisphere compared with surrounding deep seawater and sediment under the plastic chamber at site BHT.

Reviewer #1 (Remarks to the Author):

I've reviewed this MS before; my initial questions regarding the validity of the degradation rate determination has been addressed, mostly sufficiently. I'd urge the authors, however, to more clearly distinguish between what they measured (eg change in film thickness / molecular properties / surface structure of the polymers, and the presence of certain genes/organisms) and what they deduce from this (eg that enzymes are released and that the plastic is mineralised. I do acknowledge that this seems logical but still a clear somatic differentiation between observation and interpretation should be made.

I also have a number of more specific points:

Abs.

L62 unclear what 'harbour plastispheres' means. Sites where plastic pollution was found? This should be rephrased. Also, the term 'plastisphere' is used a lot but it is misleading as 'sphere' is indicating the scale of 'ecosphere', 'hydrosphere', 'geosphere' etc.

Intro

L68 this sentence is missing syntax, I suggest to change this to:... a large proportion of ~8 Mt/year ends up in the marine environment, e.g. through riverine transport.

R&D

L130ff This is mostly a description of the graphs, which is ok, but it also does contain parts that are assumed but not measured (I give only two examples here: 'the biofilm is releasing enzymes to degrade the substrate'. That seems logical but is not measured and this should be made clear. Similarly, whether water soluble degradation products are released and whether these are further mineralised is also not clear). This is discussed in a better way further in the MS, eg Line 354 ff where the authors rightly state that this data is suggesting the involvement of enzymes in biodegradation.

Reviewer #2 (Remarks to the Author):

All comments have been addressed

Point by Point Reply to Reviewer #1

I've reviewed this MS before; my initial questions regarding the validity of the degradation rate determination has been addressed, mostly sufficiently. I'd urge the authors, however, to more clearly distinguish between what they measured (eg change in film thickness / molecular properties / surface structure of the polymers, and the presence of certain genes/organisms) and what they deduce from this (eg that enzymes are released and that the plastic is mineralised. I do acknowledge that this seems logical but still a clear somatic differentiation between observation and interpretation should be made.

(Reply) We thank the reviewer for his/her time in reviewing our manuscript. We are glad that our manuscript has addressed your concerns. We also appreciate the further comments that you have provided, which we have taken into consideration in this revised manuscript.

With regards to distinguishing the experimental observation and the possible mechanism of degradation, (particularly in connection with Fig. 2) we have added few extra text (Line 177-181) to clarify what we have observed and its subsequent interpretation.

Line 177-181: However, in this study, no further experiments were conducted to directly support that microorganisms actually secrete extracellular depolymerases in the extreme environment of the deep sea. Subsequently, it is unclear whether the microorganisms at deep sea completely degrade biodegradable plastics into carbon dioxide and water through water soluble intermediates.

(Comment)

Abs.

L62 unclear what 'harbour plastispheres' means. Sites where plastic pollution was found? This should be rephrased. Also, the term 'plastisphere' is used a lot but it is misleading as 'sphere' is indicating the scale of 'ecosphere', 'hydrosphere', 'geosphere' etc.

(Reply) We apologize for the lack of clarity in the statement at L62. The abstract had to be completely rewritten due to editorial requirements. Therefore, the sentence "... harbor plastispheres" has been removed from the abstract.

Regarding "plastispheres", the No. 32 reference states that "plastispheres" are defined as microbial communities formed on plastic debris in the ocean. We have clearly stated this at the beginning of the Microbial Community Analysis of Plastispheres section in the text (Line 283).

(Comment)

Intro

L68 this sentence is missing syntax, I suggest to change this to:... a large proportion of ~8 Mt/year ends up in the marine environment, e.g. through riverine transport.

(Reply) We thank the reviewer for his/her suggestion. We have modified the sentence as follows:

Line 56-58: Plastic products should be collected and recycled after use; however, it has been reported that approximately 8 million tons of plastic waste ends up in the marine environment through the rivers every year¹⁻⁴.

(Comment)

R&D

L130ff This is mostly a description of the graphs, which is ok, but it also does contain partst that are assumed but not measured (I give only two example here: ‘the biofilm is releasing enzymes to degrade the substrate’. That seems logical but is not measured and this should be made clear. Similarly, whether water soluble degradation products are released and whether these are further mineralised is also not clear). This is discussed in a better way further in the MS, eg Line 354 ff where the authors rightly state that this data is suggesting the involvement of enzymes in biodegradation.

(Reply) We thank the reviewer for his/her suggestion. As the reviewer mentioned, there is no direct evidence to support that microorganisms actually secrete extracellular depolymerases in the extreme environment of the deep sea and completely degrade biodegradable plastics into carbon dioxide and water forming water soluble intermediates. Accordingly, to eliminate any misunderstanding, the following sentence was added at the end of explanation of Fig.2 i and 2j.

Line 177-181: However, in this study, no further experiments were conducted to directly support that microorganisms actually secrete extracellular depolymerases in the extreme environment of the deep sea. Subsequently, it is unclear whether the microorganisms at deep sea completely degrade biodegradable plastics into carbon dioxide and water through water soluble intermediates.